# Context-Aware Patch Representations for Multiple Instance Learning

**Andreas Lolos**[*1,2] ⓘD                                    ANDREASLOLOS@PHYS.UOA.GR
**Theofilos Christodoulou**[*2]                           TH.CHRISTODOULOU@ATHENARC.GR
**Aris L. Moustakas**[1,2]                                              ARISLM@PHYS.UOA.GR
**Stergios Christodoulidis**[3,4]        STERGIOS.CHRISTODOULIDIS@CENTRALESUPELEC.FR
**Maria Vakalopoulou**[2,3,4]              MARIA.VAKALOPOULOU@CENTRALESUPELEC.FR

[1] *National and Kapodistrian University of Athens, Greece*

[2] *Archimedes, Athena Research Center, Greece*

[3] *MICS Laboratory, CentraleSupélec, Université Paris-Saclay*

[4] *IHU PRISM, National Center for Precision Medicine in Oncology, Gustave Roussy*

**Editors:** Accepted for publication at MIDL 2026

## Abstract

In computational pathology, weak supervision has become the standard for deep learning due to the gigapixel scale of WSIs and the scarcity of pixel-level annotations, with Multiple Instance Learning (MIL) established as the principal framework for slide-level model training. In this paper, we introduce CAPRMIL, a novel setting for MIL methods, inspired by advances in Neural Partial Differential Equation (PDE) solvers. Instead of relying on complex attention-based aggregation, we propose an efficient, aggregator-agnostic framework that removes the complexity of correlation learning from the MIL aggregator. CAPRMIL produces rich context-aware patch embeddings that promote effective correlation learning on downstream tasks. By projecting patch features —extracted using a frozen patch encoder— into a small set of global context/morphology-aware tokens and utilizing multi-head self-attention, CAPRMIL injects global context with linear computational complexity with respect to the bag size. Paired with a simple Mean MIL aggregator, CAPRMIL matches state-of-the-art (SOTA) slide-level performance across multiple public pathology benchmarks, while reducing the total number of trainable parameters by 48%–92.8% versus SOTA MILs, lowering FLOPs during inference by 52%–99%, and ranking among the best models on GPU memory efficiency and training time. Our results indicate that learning rich, context-aware instance representations before aggregation is an effective and scalable alternative to complex pooling for whole-slide analysis. Our code is available at: https://github.com/mandlos/CAPRMIL

**Keywords:** Digital Pathology, Multiple Instance Learning, Context Aware Representations

## 1. Introduction

Whole Slide Image (WSI) analysis has become the foundation of clinical practice in computational pathology (Alkhalaf et al., 2024; Wang et al., 2024), however their sheer size poses a significant challenge for Deep Learning approaches (Brixtel et al., 2022; Lu et al.,

---

[*] Contributed equally

2021b; Gadermayr and Tschuchnig, 2024). At the same time, pixel-level annotations are prohibitively expensive and time-consuming, resulting in clinical datasets that typically provide only slide-level labels rather than fine-grained annotations (Lu et al., 2021b; Song et al., 2023; Gadermayr and Tschuchnig, 2024).

To address the computationally prohibitive size of WSIs and the lack of pixel-level annotations, Multiple Instance Learning (MIL) has been established as the standard framework for WSI analysis. The MIL pipeline comprises patch feature extraction, typically adopting pre-trained foundation models (Xiong et al., 2025), followed by aggregation/pooling to produce the slide-level representation for downstream tasks. In recent years, attention-based mechanisms have emerged as a promising approach for a trainable MIL aggregator (Ilse et al., 2018; Wang et al., 2024; Gadermayr and Tschuchnig, 2024), due to their impressive correlation learning capabilities. While effective, approaches that utilize standard attention directly on the patch embeddings face computational bottlenecks due to the quadratic complexity of the attention operator (Shao et al., 2021). Attention-based MIL methods for WSI have also been found to be highly susceptible to overfitting and offer limited interpretability (Zhang et al., 2025b), while often lacking principled uncertainty quantification (Sun et al., 2025; Cui et al., 2023; Lolos et al., 2026), limiting the potential of clinical translation. Therefore, developing aggregation strategies that can effectively model instance interactions, handle the challenges inherent to long sequence processing in WSIs, and provide reliable representations remains an active area of research (Bilal et al., 2023; Fang et al., 2024).

At the same time, we identify that neural Partial Differential Equation (PDE) solvers (Li et al., 2020; Hao et al., 2023; Wu et al., 2024) face a similar challenge: how to achieve efficient and reliable correlation learning in large-scale inputs. Solving PDEs often includes modeling complex phenomena that may cause long-distance interactions, on domains discretized into millions of mesh points (Grossmann et al., 2024). Attention-based methods have been used in PDE modeling, but they also face prohibitive computational cost and degraded correlation learning due to the large scale of the input (Katharopoulos et al., 2020; Wu et al., 2024). Therefore, we argue that ideas that have successfully tackled these problems in the domain of Surrogate PDE solvers could provide new insights in digital pathology.

In this work, we introduce CAPRMIL, a novel and efficient attention-based MIL framework for WSI analysis, proposing a paradigm shift by removing the complexity of correlation learning from the MIL aggregator, using context-aware patch representations. Following the architecture of Transolver (Wu et al., 2024; Luo et al., 2025), which shows promising results in efficient PDE modeling, we leverage Multi-Head Self-Attention (MSA) over a small set of global context-aware tokens, achieving linear computational complexity with respect to the input and promoting effective correlation learning on downstream tasks. More precisely, our main contributions are summarized as follows:

1. **We propose a novel and efficient MIL setting based on the Transolver architecture.** Tackling the challenge of the large dimensionality of the input, CAPRMIL introduces a bottleneck before the attention operator, which consists of: (1) soft clustering of the patch embeddings and (2) aggregating each cluster into a context-aware token. By utilizing MSA over the context-aware tokens, CAPRMIL achieves linear computational complexity with respect to the bag size and produces rich morphology/ context-aware patch representations.

2. **A highly parameter-efficient formulation.** Our approach performs on par with current state-of-the-art MIL heads, while reducing the total number of trainable parameters by 48% compared to ABMIL and up to 92.8% compared to SOTA transformer-based MILs. This significantly reduces the computational requirements during training and inference in terms of time, FLOPS, and memory utilization.

3. **A scalable, aggregator-agnostic formulation that can be adapted in multiple MIL heads**. Our formulation is independent of the MIL aggregator, and it can be applied in different commonly used MIL settings with small computational overhead.

We evaluate CAPRMIL on various publicly available computational pathology datasets. Paired with a simple MeanMIL aggregator, our method matches SOTA performance, while achieving leading efficiency, highlighting a highly efficient and adaptable MIL framework.

## 2. Related Work

**MIL-based frameworks for digital pathology.** During the last years, many different MIL settings have been introduced and extensively tested in different settings (Shao et al., 2025). Depending on the mechanism of aggregation that they are using, they can be grouped into different categories. Among the most popular attention-based methods, we can note ABMIL (Ilse et al., 2018), CLAM (Lu et al., 2021b) and DSMIL (Li et al., 2021). Moreover, TransMIL (Shao et al., 2021) was among the first to introduce a transformer network specifically for WSI, in order to model both morphological and spatial correlations. Building on top of this, DGRMIL (Zhu et al., 2025) utilizes a set of learnable "global vectors" to summarize distinct morphological patterns and computes cross-attention between the instances and these global vectors, effectively achieving linear scaling. Finally, probabilistic-based MIL methods, such as BayesMIL (Cui et al., 2023), argue that standard attention scores are unreliable proxies for interpretability and address this by introducing a probabilistic instance-wise attention module that yields patch-level uncertainty estimates. Similarly, SGPMIL (Lolos et al., 2026) targets the lack of uncertainty estimation in deterministic models by learning a posterior distribution over attention scores, using an input independent inducing set of prototypes.

**Attention-based Neural PDE solvers.** Solving Partial Differential Equations (PDEs) is fundamental to modeling complex phenomena in science and engineering. While traditional numerical approaches such as the Finite Element Method (FEM) offer high accuracy, they typically require discretization of the domain into high-resolution meshes —often containing millions of mesh points— resulting in prohibitive computational costs (Grossmann et al., 2024). Consequently, deep learning-based neural operators have emerged as efficient surrogates, capable of learning the mapping between model state and solution fields directly from data (Li et al., 2020; Lu et al., 2021a; Wu et al., 2024). Transformer architectures have been increasingly utilized in neural PDE solvers due to their ability to model global dependencies (Li et al., 2022). However, they often face computational bottlenecks due to the quadratic complexity of standard self-attention (Katharopoulos et al., 2020; Luo et al., 2025). Furthermore, simply applying attention to individual mesh points may fail to capture the intricate high-order physical correlations governing the system, as the model can be-

come overwhelmed by low-level geometric details, thus preventing effective relation learning (Wu et al., 2022). We identify that challenges inherent to long-sequence processing, such as computational complexity and efficient correlation learning, are common in both large-scale physical simulations and WSI analysis. Surprisingly, to the best of our knowledge, the use of neural PDE solvers has not been explored in digital pathology.

**The Transolver Architecture.** To address the prohibitive computational cost and degraded correlation learning due to the large size of the input, the Transolver architecture was introduced as a Transformer-based PDE solver for general geometries (Wu et al., 2024), and later scaled in larger settings (Luo et al., 2025). Their architecture introduces Physics-Attention, proposing that a domain discretized to $N$ mesh points can be decomposed into a set of $M \ll N$ physically consistent clusters ("slices"), which can then be aggregated into "physics-aware tokens", forming a compact latent representation of distinct physical states. Standard Multi-Head Self-Attention (MSA) can then be applied to these tokens for correlation modeling with complexity $O(M^2)$, achieving linear scaling with respect to the number of mesh points. By explicitly modeling "physical states" rather than individual points, the model becomes more robust to geometric variations and discretization artifacts, while the learned slices have been shown to correspond to meaningful physical regions, enhancing the model's interpretability and generalization capability (Wu et al., 2024; Luo et al., 2025). Drawing a parallel to digital pathology, both neural PDE solvers and MIL models face the fundamental challenge of efficiently learning correlations over massive sequences of instances (mesh points in PDEs, patches in WSIs). Viewed through this lens, Transolver's Physics-Attention constitutes a promising approach to facilitate efficient global correlation modeling, by projecting the high-dimensional input space onto a compact set of latent variables.

**Prototype-based Multiple Instance Learning.** ProtoMIL (Rymarczyk et al., 2022) proposes a self-explainable MIL framework that learns a fixed set of trainable prototype vectors and represents each slide by aggregating, via attention pooling, the maximum L2-similarity between each prototype and its most similar patch embedding, enabling case-based reasoning through explicit prototype–patch matches. However, each prototype is driven by a single maximally activating patch, and the semantic meaning of the learned prototypes is not exhaustively validated. TPMIL (Yang et al., 2023) refines instance-level features by softly assigning all patch embeddings to trainable prototypes using attention-derived pseudo-labels to better capture intra-class morphological heterogeneity, but this refinement is tightly coupled to a specific attention-based MIL aggregator, limiting architectural flexibility. PAMIL (Liu et al., 2024) similarly employs cross-attention between learnable prototypes and patch embeddings to jointly aggregate instance-level and prototype-level information for slide classification, with prototype refinement again tied to a fixed attention-based aggregator. Prototype-Based MIL (Sun et al., 2025) adopts a two-stage framework in which human-interpretable concepts are first learned via a sparse autoencoder and subsequently aggregated for slide classification using attention-weighted sum pooling and a linear classifier, making overall performance strongly dependent on the quality of the separately learned concept representations. Taken together, CAPRMIL is adjacent to these prototype-based and concept-based MIL methods in that it exploits morphological redundancy, but differs fundamentally by learning input-conditioned, softly assigned cluster

tokens that are jointly optimized in a single-stage, end-to-end manner and can be seamlessly combined with arbitrary MIL aggregation operators.

## 3. Methodology

In this work, we propose CAPRMIL, a novel and efficient MIL framework, designed to overcome the limitations of standard attention in WSI analysis. Unlike prior methods that treat patches as isolated units, CAPRMIL projects patch embeddings into morphology units via soft clustering, and aggregates them into a compact set of context-aware, low-dimensional global tokens, over which self-attention is performed. Global contextual information is then propagated back to the patch embeddings via context broadcasting. By attending to the tokens rather than patch embeddings, CAPRMIL achieves linear scaling with respect to the bag size, while maintaining strong representational capacity and high parameter efficiency.

### 3.1. Model Architecture

The CAPRMIL framework for WSI consists of three sequential stages (Figure 1): (1) an initial projection of WSI patches into patch embeddings using a pre-trained encoder as frozen backbone, (2) a stack of CAPRMIL Blocks that use multi-head self-attention over global token representations to produce context/morphology-aware patch embeddings, and (3) a final MIL aggregation and classification head to produce the slide-level prediction.

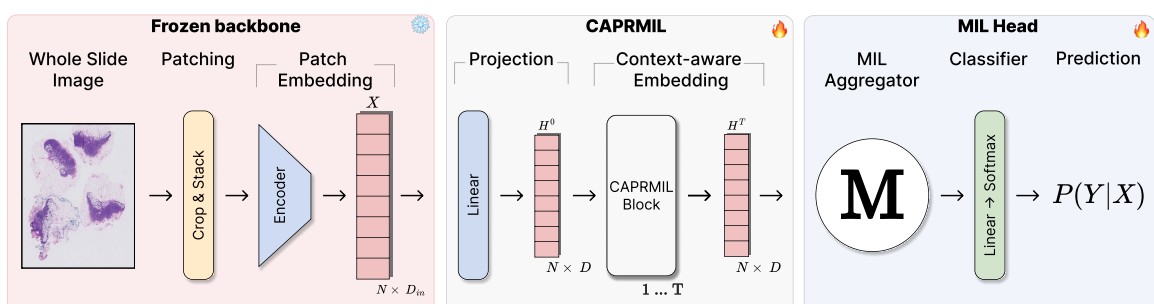

Figure 1: CAPRMIL Framework. The WSI is tessellated into patches which are encoded into patch embeddings using a frozen backbone. After a linear projection, $T$ consecutive CAPRMIL Blocks augment global context to yield context-aware patch embeddings. A MIL aggregator and classifier then produce the final slide-level prediction.

#### 3.1.1. FEATURE PROJECTION

A batch of WSIs is represented as a batch of bags $X \in \mathbb{R}^{B \times N \times D_{in}}$ of $N$ patch embeddings of dimension $D_{in}$, with batch size $B$. These embeddings are projected into a latent space of dimension $D \ll D_{in}$ via a learnable linear layer followed by Layer Normalization (LN), GELU activation, and Dropout, yielding patch representations $\mathbf{H}^{(0)}$ as input to the first CAPRMIL Block:

$$\mathbf{H}^{(0)} = \text{Dropout}(\text{GELU}(\text{LN}(\text{Linear}(\mathbf{X})))) \in \mathbb{R}^{B \times N \times D}$$

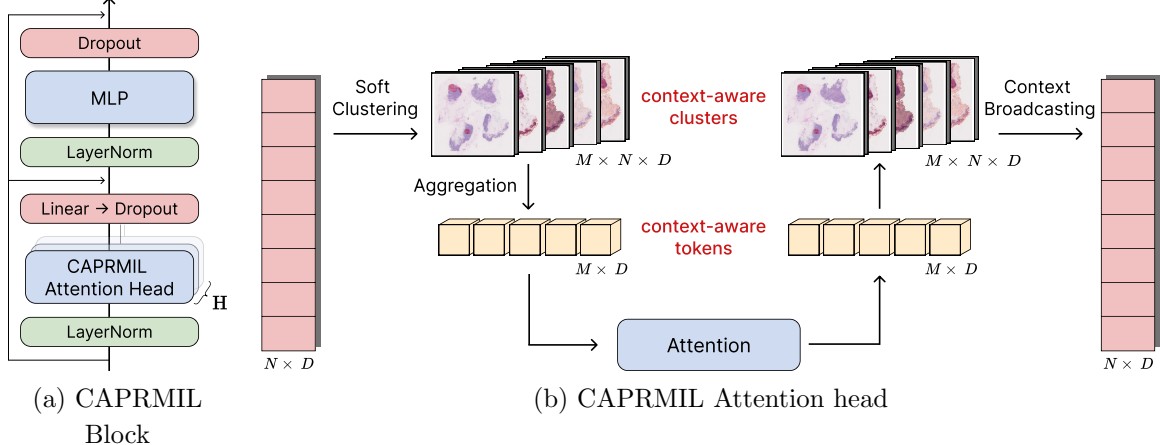

(a) CAPRMIL Block

(b) CAPRMIL Attention head

Figure 2: (a) The CAPRMIL Block follows a transformer encoder architecture with multihead self-attention. Each CAPRMIL Block contains $H$ CAPRMIL Attention heads and their outputs are concatenated. Skip connections are implemented after every Dropout. (b) The CAPRMIL Attention head projects patch embeddings into $M$ clusters, via soft clustering. Each cluster is aggregated into a context-aware token and attention is applied to the set of $M$ tokens. The refined tokens are projected back to the input latent space via context broadcasting.

### 3.1.2. The CAPRMIL Block

To capture high-order correlations without the quadratic cost of standard self-attention, the CAPRMIL Block adopts the Transolver architecture, performing attention over low-dimensional, context-aware global tokens to achieve linear complexity with respect to the bag size. As illustrated in Figure 2a, it follows a Transformer encoder-style design with $H$ CAPRMIL heads and shared projection matrices across heads, augmenting patch embeddings with global context to produce rich, morphology-aware representations, formulated as:

$$\mathbf{H}' = \mathbf{H}^{(l-1)} + \text{Dropout}(\text{CAPRMIL Attention}(\text{LN}(\mathbf{H}^{(l-1)})))$$

$$\mathbf{H}^{(l)} = \mathbf{H}' + \text{Dropout}(\text{MLP}(\text{LN}(\mathbf{H}')))$$

for $l \in [1, T]$, for $T$ consecutive CAPRMIL Blocks. The MLP comprises two linear layers with GELU activation. CAPRMIL attention produces context-aware patch representations by aggregating information across multiple attention heads.

### 3.1.3. CAPRMIL Attention head

CAPRMIL adopts the Physics-Attention mechanism from Transolver to enable efficient correlation learning on large-scale inputs. As illustrated in Figure 2b, it operates in four stages: (1) soft clustering of patch representations into morphology-aware clusters, (2) aggregation into morphology-aware tokens, (3) self-attention over these tokens, and (4) broadcasting the refined tokens back to the input space to produce context-aware patch representations.

**Soft Clustering.** Intuitively, this step compresses the $N$ patch embeddings into a compact set of $M \ll N$ morphology-aware tokens using soft assignment. Let $\mathbf{H} \in \mathbb{R}^{B \times N \times D}$ denote the input patch representations. $\mathbf{H}$ is mapped via two learnable projections into $\mathbf{x}, \mathbf{f} \in \mathbb{R}^{B \times N \times (HD_{\text{head}})}$, $D_{\text{head}} = D/H$, by linear layers $\mathbf{W}_x, \mathbf{W}_f \in \mathbb{R}^{D \times (HD_{\text{head}})}$ and reshaped into $\tilde{\mathbf{x}}, \tilde{\mathbf{f}} \in \mathbb{R}^{B \times H \times N \times D_{\text{head}}}$. All $N$ patch embeddings are then softly assigned to $M$ context-aware clusters per head. A learnable projection $\mathbf{W}_{\text{cluster}} \in \mathbb{R}^{D_{\text{head}} \times M}$, initialized orthogonally, produces:

$$\mathbf{W} = \text{Softmax}_m \left( \frac{\tilde{\mathbf{x}} \mathbf{W}_{\text{cluster}}}{\tau} \right)$$

The softmax is applied along the cluster dimension, such that the assignment weights form a valid patch-to-cluster probability score, meaning that $\sum_{m=1}^{M} W_{b,h,n,m} = 1$. Here, $\mathbf{W} \in \mathbb{R}^{B \times H \times N \times M}$ denotes the soft assignment matrix. The number of clusters is controlled by $M$, while the learnable positive temperature $\tau \in \mathbb{R}_+^H$ regulates the assignment entropy of each attention head. Cluster-specific tokens $\mathbf{S} \in \mathbb{R}^{B \times H \times M \times D_{\text{head}}}$ are then computed as weighted combinations of the input embeddings:

$$\mathbf{S}_{b,h,m,d} = \frac{\sum_{n=1}^{N} W_{b,h,n,m} \, \tilde{\mathbf{f}}_{b,h,n,d}}{\sum_{n=1}^{N} W_{b,h,n,m} + \varepsilon}$$

**Self-Attention.** To achieve linear scaling with respect to the bag size, CAPRMIL performs self-attention over a compact set of $M \ll N$ morphology-aware tokens, obtained via soft clustering and aggregation, rather than over individual patches. Given $M$ morphology-aware tokens $\mathbf{S}$ per head, we apply Multi-Head Self-Attention. Query ($\mathbf{Q}$), Key ($\mathbf{K}$), and Value ($\mathbf{V}$) are obtained via shared linear projections $\mathbf{W}_q, \mathbf{W}_k, \mathbf{W}_v \in \mathbb{R}^{D_{head} \times D_{head}}$ of the head-wise token embeddings $\mathbf{S}$. Attention is then given by:

$$\text{Attn} = \text{Softmax}\left( \frac{\mathbf{Q}\mathbf{K}^\top}{\sqrt{D_{\text{head}}}} \right), \quad \mathbf{S}' = \text{Dropout}(\text{Attn} \cdot \mathbf{V})$$

with $\mathbf{S}' \in \mathbb{R}^{B \times H \times M \times D_{head}}$. Since $(M \ll N)$, applying the attention operator over the context-aware tokens —instead of the $N$ patch embeddings— reduces computational complexity and allows the model to scale linearly with the input. Because tokens aggregate global context, CAPRMIL learns meaningful correlations, beyond local spatial features.

**Context Broadcasting.** The refined tokens $\mathbf{S}'$ are broadcast back to the input latent space using the same assignment weights $\mathbf{W}$ from the soft clustering step, reconstructing each patch representation as a weighted combination of updated tokens:

$$\mathbf{O}_{b,h,n,d} = \sum_{m=1}^{M} \mathbf{S}'_{b,h,m,d} \mathbf{W}_{b,h,n,m}, \quad \mathbf{O} \in \mathbb{R}^{B \times H \times N \times D_{\text{head}}}.$$

Head-wise representations are concatenated into $\mathbf{H}^{(T)} \in \mathbb{R}^{B \times N \times (HD_{head})}$ and linearly projected to the model dimension, yielding the final context-aware patch representations.

### 3.1.4. Aggregation and Prediction

After $T$ CAPRMIL Blocks, the context-aware patch representations $\mathbf{H}^{(T)} \in \mathbb{R}^{B \times N \times D}$ are aggregated into a slide-level embedding $\mathbf{z} \in \mathbb{R}^{B \times 1 \times D}$ using an MIL aggregator $\mathcal{A}(\cdot)$:

$$\mathbf{z} = \mathcal{A}\big(\mathbf{H}^{(T)}\big)$$

In practice, $\mathcal{A}$ may correspond to a simple non-parametric pooling operator, such as mean or max pooling, or to more expressive attention-based or gated-attention aggregators. Importantly, these choices do not affect the structure of the CAPRMIL blocks and can be interchanged without modification; formal definitions of the aggregation functions are provided in Appendix C. A final linear classifier then maps $\mathbf{z}$ to task-specific class logits.

### 3.2. Computational Efficiency

CAPRMIL addresses the "curse of dimensionality" by decoupling the sequence length $N$ from the attention mechanism. Since the attention operator displays quadratic computational complexity, attending to all $N$ patch embeddings would yield $O(N^2)$ complexity. CAPRMIL Attention instead attends to the $M$ context-aware tokens, achieving an overall complexity of $O(MND + M^2D)$. Given that the number of tokens $M$ is a constant with $M \ll N$, the model achieves linear computational complexity with respect to the input size $N$, making it ideal to model long sequences.

## 4. Experiments & Results

**Datasets, Tasks and Evaluation Metrics.** We evaluate our approach on four WSI benchmarks: **CAMELYON16** (Ehteshami Bejnordi et al., 2017) for tumor detection, **TCGA-NSCLC** (Cooper et al., 2018; Campbell et al., 2016) for lung cancer subtyping, **BRACS** (Brancati et al., 2022) for coarse breast lesion classification, and **PANDA** (Bulten et al., 2022) for prostate ISUP grading. Dataset-specific evaluation protocols, metrics, and implementation details are provided in the Appendix. We report slide-level classification performance and calibration using area under the curve (**AUC**) and adaptive expected calibration error (**ACE**) (Nixon et al., 2019).

### 4.1. Slide-level Performance and Parameter Efficiency

| | CAMELYON16 | | TCGA-NSCLC | | PANDA | | BRACS | | Params | FLOPs |
|---|---|---|---|---|---|---|---|---|---|---|
| | AUC | ACE | AUC | ACE | $\kappa$ | ACE | AUC | ACE | (M) | (G) |
| ABMIL (Ilse et al., 2018) | $\mathbf{.987}_{\mathbf{005}}$ | $.036_{004}$ | $.973_{009}$ | $.039_{008}$ | $.910_{028}$ | $.044_{015}$ | $\underline{.852}_{025}$ | $\underline{.175}_{007}$ | .660 | 1.31 |
| CLAM (Lu et al., 2021b) | $.986_{004}$ | $.044_{027}$ | $.953_{004}$ | $.056_{016}$ | $.927_{025}$ | $.031_{018}$ | $.850_{021}$ | $.183_{011}$ | .920 | 1.84 |
| TransMIL (Shao et al., 2021) | $.978_{004}$ | $.044_{012}$ | $.970_{012}$ | $.046_{019}$ | $.911_{030}$ | $.043_{021}$ | $.826_{032}$ | $.186_{012}$ | 2.67 | 85.02 |
| DGRMIL (Zhu et al., 2025) | $.967_{018}$ | $.027_{021}$ | $.974_{011}$ | $\underline{.038}_{022}$ | $.933_{047}$ | $.036_{025}$ | $.818_{035}$ | $.186_{023}$ | 4.34 | 79.88 |
| PAMIL (Liu et al., 2024) | $.986_{003}$ | $.032_{013}$ | $.970_{015}$ | $\underline{.043}_{018}$ | $.941_{060}$ | $.026_{032}$ | $.830_{038}$ | $.193_{018}$ | .796 | 1.32 |
| BayesMIL (Cui et al., 2023) | $.975_{006}$ | $\mathbf{.023}_{\mathbf{006}}$ | $.973_{021}$ | $\mathbf{.033}_{\mathbf{017}}$ | $.926_{031}$ | $.031_{016}$ | $.829_{022}$ | $.183_{028}$ | 1.32 | 2.63 |
| SGPMIL (Lolos et al., 2026) | $\mathbf{.987}_{\mathbf{008}}$ | $.026_{009}$ | $.973_{014}$ | $.047_{027}$ | $\mathbf{.955}_{\mathbf{037}}$ | $.028_{022}$ | $\mathbf{.870}_{\mathbf{026}}$ | $\mathbf{.142}_{\mathbf{032}}$ | 1.21 | 2.44 |
| Mean | $.693_{046}$ | $.241_{022}$ | $\mathbf{.979}_{\mathbf{015}}$ | $.041_{019}$ | $.924_{028}$ | $.035_{013}$ | $.738_{006}$ | $.223_{015}$ | $\mathbf{.130}$ | $\mathbf{.260}$ |
| **CAPRMIL+Mean** | $.975_{006}$ | $.028_{006}$ | $\underline{.978}_{016}$ | $\mathbf{.033}_{\mathbf{021}}$ | $\underline{.944}_{053}$ | $\mathbf{.021}_{\mathbf{024}}$ | $.850_{031}$ | $.189_{026}$ | $\underline{.314}$ | $\underline{.628}$ |

Table 1: Slide-level performance comparison across datasets. Results are reported as AUC/$\kappa$, and ACE. FLOPs are measured per forward pass for a bag of 1000 patch embeddings at inference. Using a mean operator in the initial projection layer before classification (i.e., our approach without the block) leads to substantial performance degradation for large bag sizes, such as CAMELYON16 and BRACS.

CAPRMIL achieves competitive AUC and calibration relative to state-of-the-art MIL methods (Table 1), with performance differences consistently within one standard deviation,

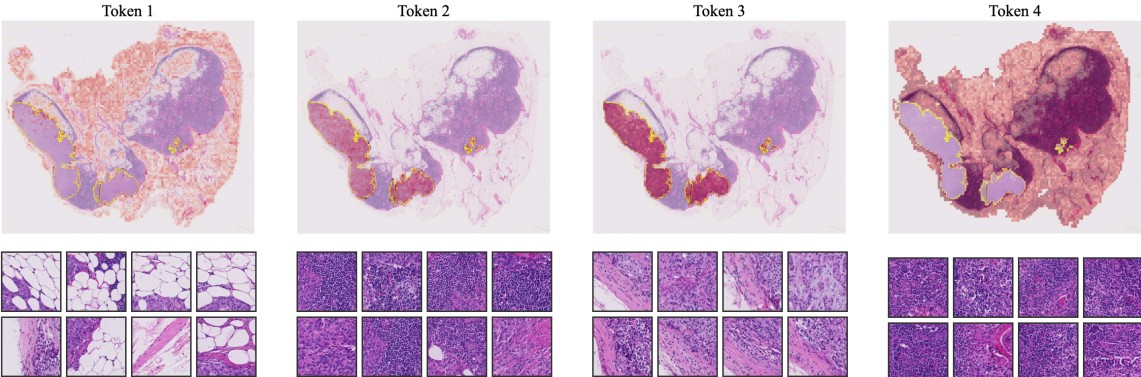

Figure 3: **Token–patch assignment heatmaps.** Test slide from CAMELYON16. *Top:* Soft assignment weights from one CAPRMIL attention head, indicating each patch's contribution to the $M$ context-aware tokens. *Bottom:* Top-8 patches per token ranked by assignment score, highlighting dominant morphological patterns for each token.

while being substantially more parameter efficient. Notably, these results are obtained using simple mean aggregation, underlining the strength of the learned context-aware patch representations. Across datasets, CAPRMIL matches parameter-efficient methods such as ABMIL, CLAM and PAMIL on CAMELYON16 and TCGA-NSCLC, and ranks among the top-performing approaches on multiclass tasks including PANDA and BRACS. At the same time, CAPRMIL reduces trainable parameters by approximately 48% relative to ABMIL, and by up to 88% and 92.8% compared to transformer-based methods such as TransMIL and DGRMIL, respectively. In terms of efficiency, CAPRMIL reduces FLOPs during inference by 52% to over 99% compared to ABMIL, TransMIL, and DGRMIL.

In contrast, naive mean aggregation —using a linear projection, mean pooling, and a linear classifier— degrades substantially on large-bag tasks. The Mean baseline underperforms by 28.2 AUC points on CAMELYON16 and 11.2 AUC points on BRACS, where bags contain 4k–20k instances. On PANDA (average bag size ∼500), mean pooling performs comparably to other methods, with a similar trend on TCGA-NSCLC. Overall, these results indicate that context-aware tokenization is critical for maintaining discriminative capacity while enabling a parameter- and computation-efficient formulation.

As seen in Figure 3 and Appendix A Figures 5–7, we observe that tokens tend to aggregate patches with visually coherent histological patterns such as adipose-rich or epithelial-dominant regions, while de-emphasizing unrelated tissue types. This observation is further supported by a cell-level analysis (Appendix D, Figures 16–18), showing that token-assigned regions exhibit distinct cellular composition profiles. The top-$k$ assigned patches per token indicate that a limited subset of instances dominates each token's construction. Specifically in Figure 3, Token 1 predominantly captures adipose-rich regions, as confirmed by their low cellular content in the top-8 assigned patches. Tokens 2 and 3 focus on tumor-related tissue, with Token 2 aggregating malignant epithelial regions, while Token 3 captures stromal or tumor-associated connective tissue. Finally, Token 4 primarily represents benign tissue, with patches exhibiting more homogeneous cellular organization.

## 4.2. Computational and Memory Efficiency

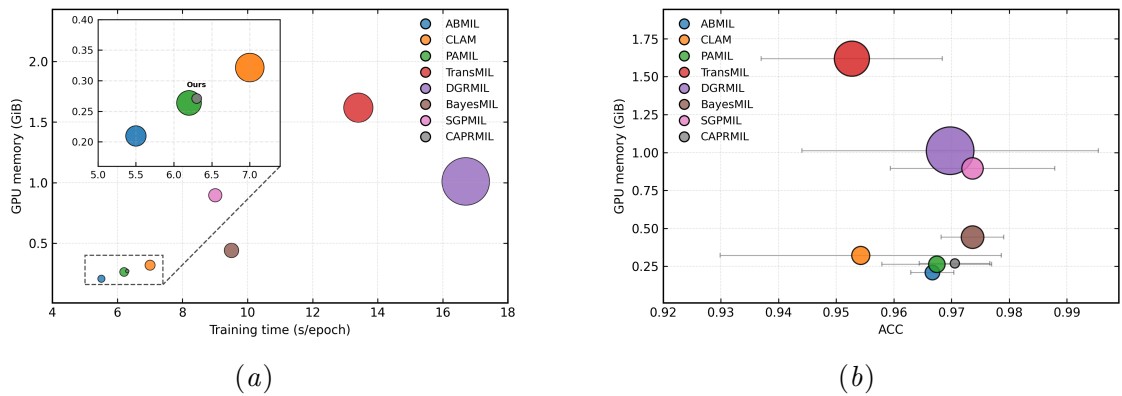

$$(a) \qquad\qquad\qquad (b)$$

Figure 4: **Model efficiency analysis**. ($a$) GPU memory footprint (peak during training, averaged over 30 epochs) vs. training time (entire training set, averaged over 30 epochs). ($b$) GPU memory footprint vs. ACC. Marker size denotes the number of trainable parameters.

While parameter count and FLOPs provide useful proxies for model efficiency, practical deployment at whole-slide scale additionally depends on empirical resource utilization. Figure 4 analyzes this by relating peak GPU memory usage, training time, and slide-level performance across competing MIL methods. As shown in Figure 4(a), CAPRMIL exhibits a substantially lower memory footprint and shorter training time compared to transformer-based approaches, reflecting its linear-scaling design and low-dimensional intermediate representations. CAPRMIL remains competitive with more computationally demanding models despite its resource-efficiency, by operating on rich, context-aware patch embeddings. Figure 4(b) further illustrates the trade-off between accuracy and memory consumption. CAPRMIL achieves high balanced accuracy, with performance differences consistently within one standard deviation of leading competitors, while operating under a significantly smaller GPU memory budget. In contrast, transformer-based methods such as TransMIL and DGRMIL incur large memory overheads for only marginal performance gains. While attention-based and probabilistic MIL methods offer stronger aggregation modules, they do so with increased computational or memory requirements, suggesting that CAPRMIL context-aware representations provide a lightweight yet competitive alternative.

### 4.3. Ablation studies

**Clusters and heads.** Varying the number of clusters $M$ while fixing $H = 8$ and the MLP ratio to 4 shows stable performance across a wide range of values ($M \in \{2, 4, 8, 16\}$), with no consistent gains from increasing the number of clusters beyond small to moderate values (Appendix Table 4, top). Similarly, increasing the number of attention heads $H$ improves performance from 2 to 8 heads but saturates thereafter, with no clear benefit on larger values (Appendix Table 4, middle). Based on these trends, we adopt $M = 4$ and $H = 8$ as balanced choices that provide sufficient contextual capacity without unnecessary complexity.

**MLP expansion ratio.** Ablating the MLP expansion ratio with fixed $M = 4$ and $H = 8$ indicates that smaller ratio slightly degrades performance, while larger ratio yields more consistent results across metrics (Appendix Table 4, bottom). We therefore use an MLP ratio of 4 in all experiments. Overall, these ablations indicate that the selected configuration ($M = 4$, $H = 8$, MLP ratio = 4) offers a robust trade-off between representational capacity and efficiency and is not sensitive to precise hyperparameter choices.

**Practical benefits vs. ABMIL in low-data regimes.** Beyond matching ABMIL under the full training set, CAPRMIL offers a clear practical advantage when labeled data are limited by shifting modeling capacity from the aggregation head to the instance representation. Table 2 compares ABMIL and CAPRMIL on the multiclass PANDA benchmark when trained on progressively smaller fractions of the training set. Across most data fractions, CAPRMIL consistently achieves lower calibration error (ACE), higher accuracy (ACC), and Cohen's $\kappa$, with the largest gains observed in the 10–50% training regimes. In particular, CAPRMIL improves ACC by 3.5%, 4.7%, and 4.3% at 10%, 25%, and 50% of the training data, respectively. For $\kappa$, the primary evaluation metric for this task, CAPRMIL outperforms ABMIL by 1.9%, 2.0%, and 2.7% in the same low-data regime, performs comparably at 75%, and again improves upon ABMIL under the full training set. In addition, CAPRMIL yields consistently lower ACE across all training fractions, indicating more reliable calibration. These results complement our efficiency analysis: while CAPRMIL introduces a modest overhead during representation construction, it reduces inference FLOPs relative to ABMIL (Table 1) and exhibits improved generalization and calibration when data are scarce. Overall, this suggests that learning context-aware patch representations prior to pooling provides a more data-efficient alternative to concentrating model capacity solely in the aggregation stage.

| Metric | Model | Fraction of training set | | | | |
| --- | --- | --- | --- | --- | --- | --- |
| | | 10% | 25% | 50% | 75% | 100% |
| ACE | ABMIL | $0.091_{0.005}$ | $0.083_{0.009}$ | $0.058_{0.017}$ | $0.043_{0.017}$ | $0.044_{0.015}$ |
| | CAPRMIL | $\mathbf{0.087_{0.005}}$ | $\mathbf{0.071_{0.008}}$ | $\mathbf{0.048_{0.017}}$ | $\mathbf{0.038_{0.021}}$ | $\mathbf{0.021_{0.024}}$ |
| ACC | ABMIL | $0.647_{0.013}$ | $0.692_{0.027}$ | $0.759_{0.053}$ | $0.818_{0.063}$ | $0.866_{0.085}$ |
| | CAPRMIL | $\mathbf{0.682_{0.012}}$ | $\mathbf{0.739_{0.030}}$ | $\mathbf{0.802_{0.046}}$ | $\mathbf{0.836_{0.073}}$ | $\mathbf{0.910_{0.079}}$ |
| $\kappa$ | ABMIL | $0.809_{0.007}$ | $0.835_{0.017}$ | $0.873_{0.032}$ | $\mathbf{0.917_{0.034}}$ | $0.910_{0.028}$ |
| | CAPRMIL | $\mathbf{0.828_{0.004}}$ | $\mathbf{0.855_{0.020}}$ | $\mathbf{0.900_{0.028}}$ | $0.915_{0.041}$ | $\mathbf{0.944_{0.053}}$ |

Table 2: Comparison of ABMIL and CAPRMIL when trained on different fractions of the labeled training set. Results are reported as mean$_\text{std}$ over 5 cross-validation folds of the PANDA dataset. Best performance for each training fraction and metric is shown in bold.

**Modularity and aggregation robustness.** Table 3 compares different MIL aggregation strategies learned together with CAPRMIL representations. In contrast to prior MIL approaches that rely heavily on sophisticated attention pooling, we observe that replacing mean aggregation with attention or gated attention leads to broadly comparable performance across datasets, within one standard deviation. Notably, on more challenging

multiclass tasks such as PANDA and BRACS, attention-based aggregators yield a performance increase from 0.8% up to 2.4% respectively, suggesting that additional aggregation capacity may be beneficial in more complex settings. Overall, these results indicate that the CAPRMIL block already encodes most of the relevant contextual and discriminative information at the patch level, rendering the choice of final aggregation largely non-critical for performance. While attention-based aggregators introduce increased parameterization, they do not provide consistent gains across all tasks, highlighting diminishing returns once strong instance representations are learned. These findings underline the modularity of CAPRMIL and demonstrate that competitive performance can be achieved with simple, parameter-efficient aggregation, while still allowing the flexibility to incorporate more expressive MIL heads when task complexity demands it.

| | CAMELYON16 | | TCGA-NSCLC | | PANDA | | BRACS | | Params |
|---|---|---|---|---|---|---|---|---|---|
| | AUC | ACE | AUC | ACE | $\kappa$ | ACE | AUC | ACE | (M) |
| **CAPRMIL+Mean** | $.975_{006}$ | $.028_{006}$ | $\mathbf{.978}_{\mathbf{016}}$ | $.033_{021}$ | $.944_{053}$ | $\underline{.021}_{024}$ | $.850_{031}$ | $.189_{026}$ | $\mathbf{.314}$ |
| **CAPRMIL+Max** | $\mathbf{.985}_{\mathbf{007}}$ | $\mathbf{.020}_{\mathbf{007}}$ | $\underline{.975}_{016}$ | $.034_{022}$ | $\mathbf{.953}_{\mathbf{042}}$ | $.023_{024}$ | $\underline{.855}_{016}$ | $.186_{024}$ | $\mathbf{.314}$ |
| **CAPRMIL+Attn** | $\underline{.977}_{004}$ | $\underline{.027}_{003}$ | $\underline{.975}_{015}$ | $\mathbf{.031}_{\mathbf{023}}$ | $.944_{046}$ | $.023_{025}$ | $.834_{032}$ | $\underline{.180}_{023}$ | $\underline{.331}$ |
| **CAPRMIL+GAttn** | $.976_{009}$ | $.033_{010}$ | $.974_{018}$ | $\underline{.032}_{020}$ | $\underline{.952}_{043}$ | $\mathbf{.019}_{\mathbf{022}}$ | $\mathbf{.874}_{\mathbf{031}}$ | $\mathbf{.171}_{\mathbf{019}}$ | $.347$ |

Table 3: Comparison of aggregation strategies within CAPRMIL. Results are reported as AUC/$\kappa$, and ACE; parameter counts include the aggregation module.

## 5. Conclusions

We present a parameter-efficient and scalable MIL framework that learns context-aware patch representations, substantially reducing reliance on complex aggregation mechanisms. Experimental results show that once rich contextual features are learned, simple pooling performs on par with more elaborate MIL heads, underscoring the robustness and modularity of the proposed approach. A current limitation is the focus on unimodal visual inputs; evaluating scalability and robustness in larger multimodal pipelines remains an interesting direction for future work.

## Acknowledgments

This work has been partially supported by project MIS 5154714 of the National Recovery and Resilience Plan Greece 2.0 funded by the European Union under the NextGenerationEU Program. Hardware resources were granted with the support of GRNET.

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

# Appendix A. Experiments

## A.1. Datasets

**CAMELYON16**   (Ehteshami Bejnordi et al., 2017) consists of 399 WSIs of sentinel lymph node tissue sections derived from women with breast cancer. The dataset is split into a training set of 270 images and a test set of 129 images. Collected from two medical centers in the Netherlands, it includes exhaustive pixel-level annotations of metastatic regions (both macrometastases and micrometastases) verified by expert pathologists. We use TRIDENT (Vaidya et al., 2025; Zhang et al., 2025a) to segment and patch the WSIs at 10× magnification (Mammadov et al., 2025) into 224×224 non-overlapping patches and utilize the UNIv1 (Chen et al., 2024) encoder for feature extraction. Similarly to Lu et al. (Lu et al., 2021b), we follow a 10-fold cross-validation protocol and report mean bag-level performance.

**TCGA-NSCLC**   We use the dataset from The Cancer Genome Atlas (TCGA) program for the non-small cell lung carcinoma (NSCLC) subtyping task (Cooper et al., 2018; Campbell et al., 2016). The dataset consists of Hematoxylin and Eosin (H&E) stained WSIs in 2 distinct cohorts: Lung Adenocarcinoma (TCGA-LUAD) and Lung Squamous Cell Carcinoma (TCGA-LUSC) (Campbell et al., 2016; Cooper et al., 2018). Specifically, we use 494 LUAD and 512 LUSC cases for a total of 1,006 slides, segment and patch at 10× magnification (Mammadov et al., 2025) into 224×224 non-overlapping patches and use the UNIv1 (Chen et al., 2024) encoder for feature extraction. Performance is reported over 4 folds.

**PANDA**   (Bulten et al., 2022) is derived from the MICCAI 2020 Prostate Cancer Grade Assessment challenge and comprises 10,609 WSIs from prostate core needle biopsies annotated, providing slide-level Gleason scores and ISUP grades alongside expert tissue annotations. We address ISUP grading (0-5) as a 6-class classification task and follow a 5-fold cross-validation protocol using stratified splits, with each fold containing approximately 80% of samples for training, 5% for validation and 15% for testing. We segment the WSIs into non-overlapping patches of size 224×224 pixels at 20× magnification (Song et al., 2024) and use the UNIv1 (Chen et al., 2024) encoder for feature extraction.

**BRACS**   (Brancati et al., 2022) dataset comprises 547 H&E stained WSIs and over 4,500 annotated regions of interest derived from 189 patients, designed to advance the automatic detection of challenging "atypical" (precancerous) lesions that are often underrepresented in other public datasets. It is annotated into seven histological subtypes, grouped into three main categories: Benign (Normal, Pathological Benign, Usual Ductal Hyperplasia), Atypical (Flat Epithelial Atypia, Atypical Ductal Hyperplasia), and Malignant (Ductal Carcinoma in Situ, Invasive Carcinoma). We specifically focus on coarse classification into the three main categories (3-class classification), using the train/validation/test split provided with the dataset. We segment the WSIs into non-overlapping patches of size 224×224 pixels at 20× magnification (Song et al., 2024), and use the UNIv1 (Chen et al., 2024) encoder for feature extraction. Performance is reported over 5 seeds.

### A.2. Implementation Details

All models are trained and evaluated in Python with PyTorch, using the same PyTorch Lightning training pipeline with identical data loading, batching, and hardware configurations. For CAPRMIL, training is performed using the standard cross-entropy loss on slide-level labels, while competing methods are trained using the loss functions specified in their original works.

The models are trained for a maximum of 30 epochs on a single A100 GPU, using full-precision (FP32) arithmetic, except MeanMIL which is trained for a maximum of 50 epochs to obtain convergence. CAPRMIL optimization employs AdamW with a base learning rate of $2 \times 10^{-4}$, weight decay of $1 \times 10^{-5}$, and momentum parameter 0.9. We use a cosine annealing learning-rate schedule, with a 6-epoch warm-up phase starting at $1 \times 10^{-5}$, and minimum learning rate of $1 \times 10^{-7}$. Early stopping was governed by a patience of 20 epochs and a performance threshold of $10^{-4}$. Learning-rate dynamics were logged every epoch, with explicit tracking of weight-decay values to enable fine-grained monitoring of the training process.

Model-specific hyperparameters and optimizer choices for competitors follow the respective original papers and are selected to ensure stable convergence based on observed loss curves. FLOPs are reported for a single forward pass during evaluation and a dummy input bag of 1000 patch embeddings and serve as a proxy for algorithmic complexity. Wall-clock training and inference times measure end-to-end execution. We additionally report peak GPU utilization as an implementation-level efficiency metric reflecting how effectively each model translates computation into hardware usage under identical experimental conditions. Complete code and instructions are publicly available at https://github.com/mandlos/CAPRMIL.

### A.3. Evaluation of context-aware tokens

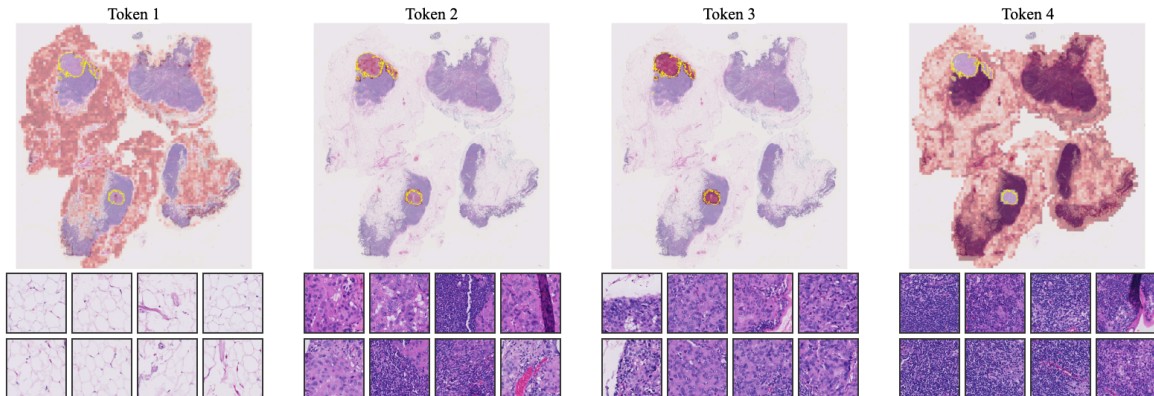

Figure 5: **Token–patch assignment heatmaps.** Test slide from CAMELYON16. *Top:* Soft assignment weights from one CAPRMIL attention head, indicating each patch's contribution to the $M$ context-aware tokens. *Bottom:* Top-8 patches per token ranked by assignment score, highlighting the dominant morphological patterns contributing to each token.

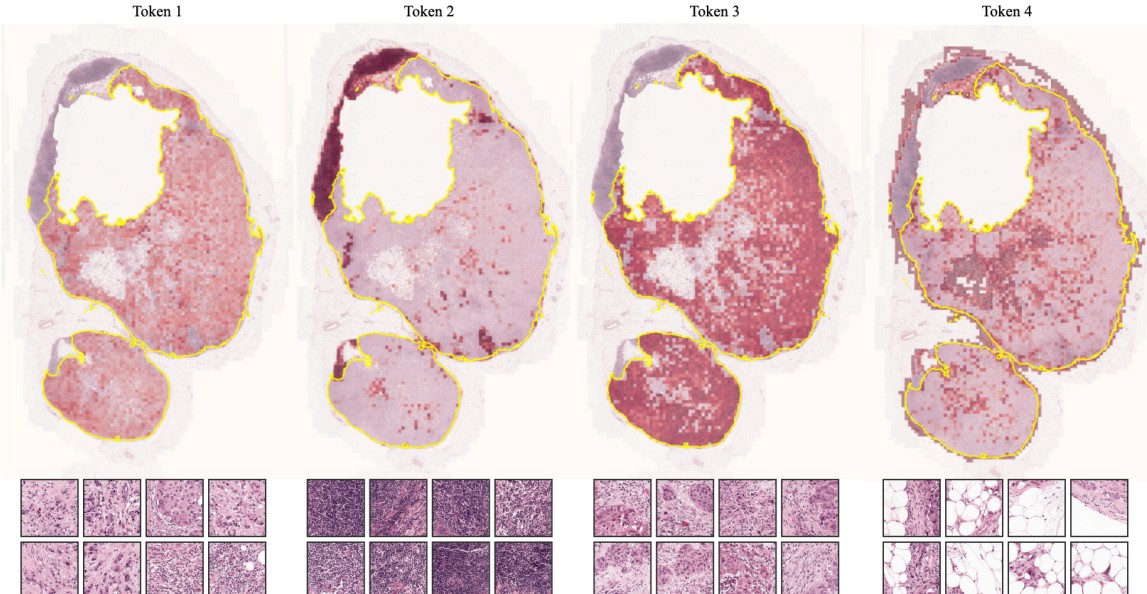

Figure 6: **Token–patch assignment heatmaps.** Test slide from CAMELYON16. *Top:* Soft assignment weights from one CAPRMIL attention head, indicating each patch's contribution to the $M$ context-aware tokens. *Bottom:* Top-8 patches per token ranked by assignment score, highlighting the dominant morphological patterns contributing to each token.

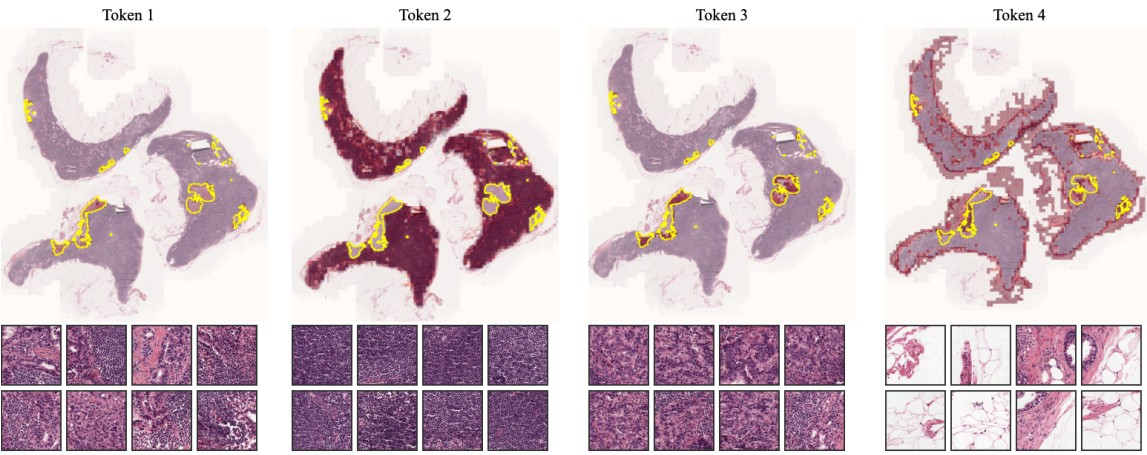

Figure 7: **Token–patch assignment heatmaps.** Test slide from CAMELYON16. *Top:* Soft assignment weights from one CAPRMIL attention head, indicating each patch's contribution to the $M$ context-aware tokens. *Bottom:* Top-8 patches per token ranked by assignment score, highlighting the dominant morphological patterns contributing to each token.

| Clusters (M) | Heads (H) | MLP ratio | AUC | ACE | Params (M) |
|---|---|---|---|---|---|
| 2 | 8 | 4 | $.971_{.009}$ | $\mathbf{.027_{.007}}$ | 0.314 |
| 4 | 8 | 4 | $\mathbf{.975_{.006}}$ | $.028_{.006}$ | 0.314 |
| 8 | 8 | 4 | $.974_{.009}$ | $.030_{.008}$ | 0.314 |
| 16 | 8 | 4 | $.973_{.008}$ | $\mathbf{.027_{.008}}$ | 0.314 |
| 4 | 2 | 4 | $.971_{.010}$ | $.031_{.011}$ | 0.326 |
| 4 | 4 | 4 | $.972_{.009}$ | $.033_{.011}$ | 0.316 |
| 4 | 8 | 4 | $\mathbf{.975_{.006}}$ | $\mathbf{.028_{.006}}$ | 0.314 |
| 4 | 12 | 4 | $.972_{.009}$ | $.032_{.006}$ | $\mathbf{0.310}$ |
| 4 | 8 | 1 | $.973_{.008}$ | $\mathbf{.027_{.006}}$ | $\mathbf{0.215}$ |
| 4 | 8 | 2 | $.969_{.011}$ | $.030_{.008}$ | 0.248 |
| 4 | 8 | 4 | $\mathbf{.975_{.006}}$ | $.028_{.006}$ | 0.314 |

Table 4: Ablation of the number of clusters $M$, attention heads $H$, and the MLP expansion ratio in the CAPRMIL block. Exhaustive ablation results for the number of clusters, attention heads, and input embedding dimensionality are provided in Appendix Figures 8, 9, 10, 11, and 12.

## Appendix B. Ablation Studies

We ablate key architectural choices of the Transolver block along three orthogonal axes: the number of clusters used for tokenization $(M)$, the number of attention heads $(H)$, and the MLP expansion ratio. Experiments are conducted on CAMELYON16, with full sweeps reported in Figures A 8, 9, 10, 11 and 12.

| Model | Training (s) | Inference (s) | Params (M) | FLOPs (G) |
|---|---|---|---|---|
| ABMIL (Ilse et al., 2018) | $\mathbf{5.5}$ | $\mathbf{0.8}$ | 0.660 | 1.31 |
| CLAM (Lu et al., 2021b) | 7.0 | 0.9 | 0.920 | 1.84 |
| TransMIL (Shao et al., 2021) | 13.4 | 1.2 | 2.67 | 85.02 |
| DGRMIL (Zhu et al., 2025) | 16.7 | 1.5 | 4.34 | 79.88 |
| PAMIL (Liu et al., 2024) | 6.2 | 0.9 | 0.796 | 1.32 |
| BayesMIL (Cui et al., 2023) | 9.5 | 1.1 | 1.32 | 2.63 |
| SGPMIL (Lolos et al., 2026) | 9.0 | 1.0 | 1.21 | 2.43 |
| CAPRMIL | 6.3 | $\mathbf{0.8}$ | $\mathbf{0.314}$ | $\mathbf{0.628}$ |

Table 5: Training and inference times (in seconds) and model sizes (number of trainable parameters in millions, M). Training times are averaged over 30 epochs, while inference times correspond to processing the full test set of 129 slides.

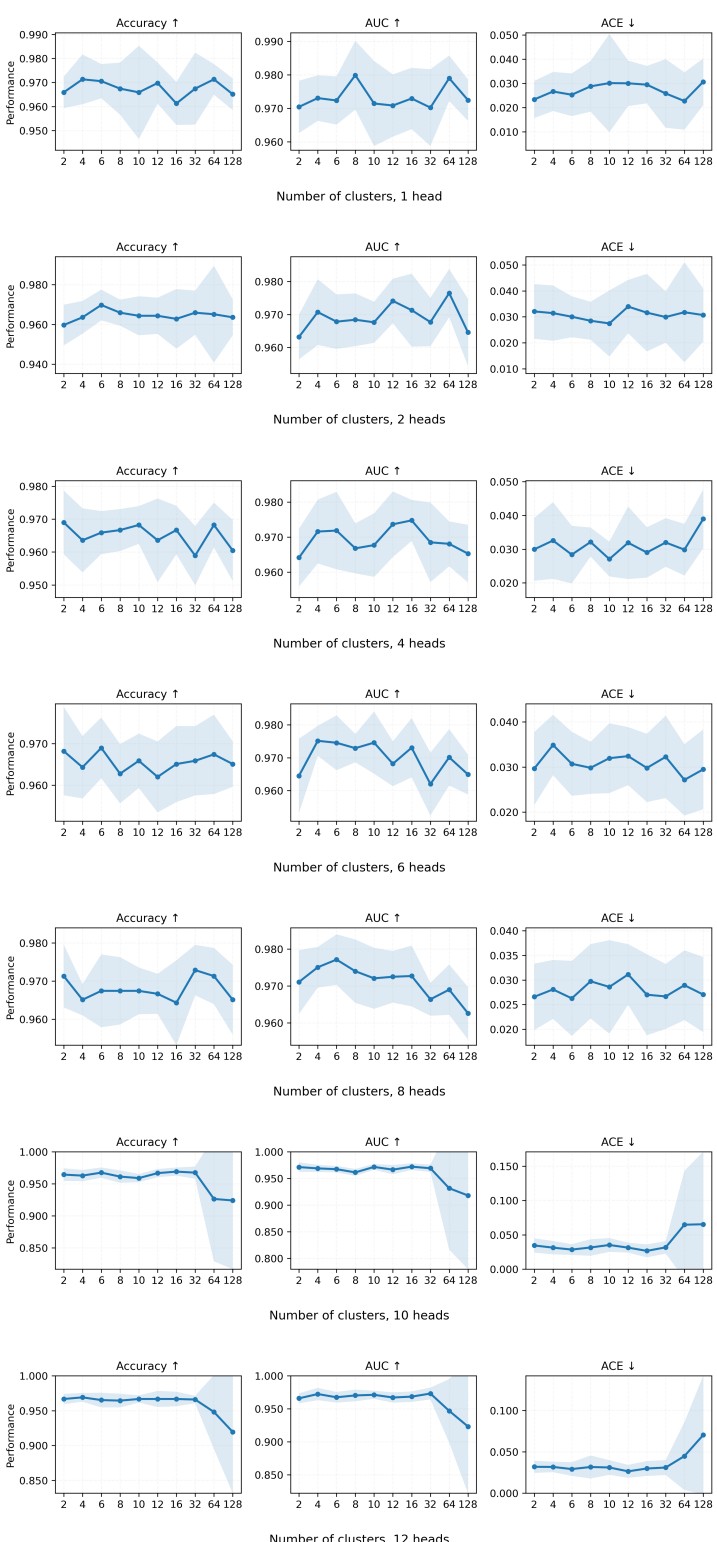

Figure 8: Ablation on the number of clusters for CAMELYON16. For each row, the number of attention heads is fixed while the number of clusters is varied. From top to bottom: 1, 2, 4, 6, 8, 10 and 12 heads.

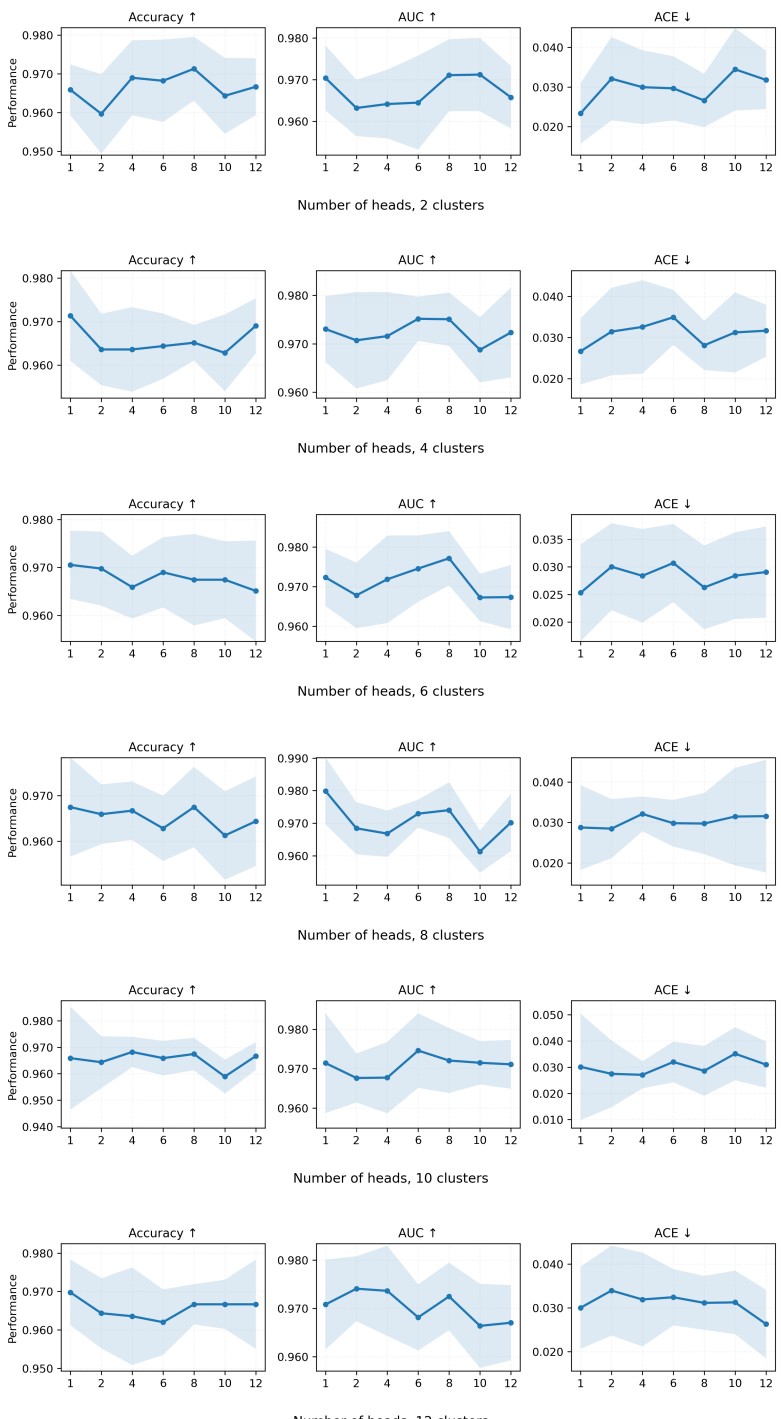

Figure 9: Ablation on the number of attention heads for a single CAPRMIL block on CAMELYON16. For each row, the number of clusters is fixed while the number of heads is varied. From top to bottom: 2, 4, 6, 8, 10, and 12 clusters.

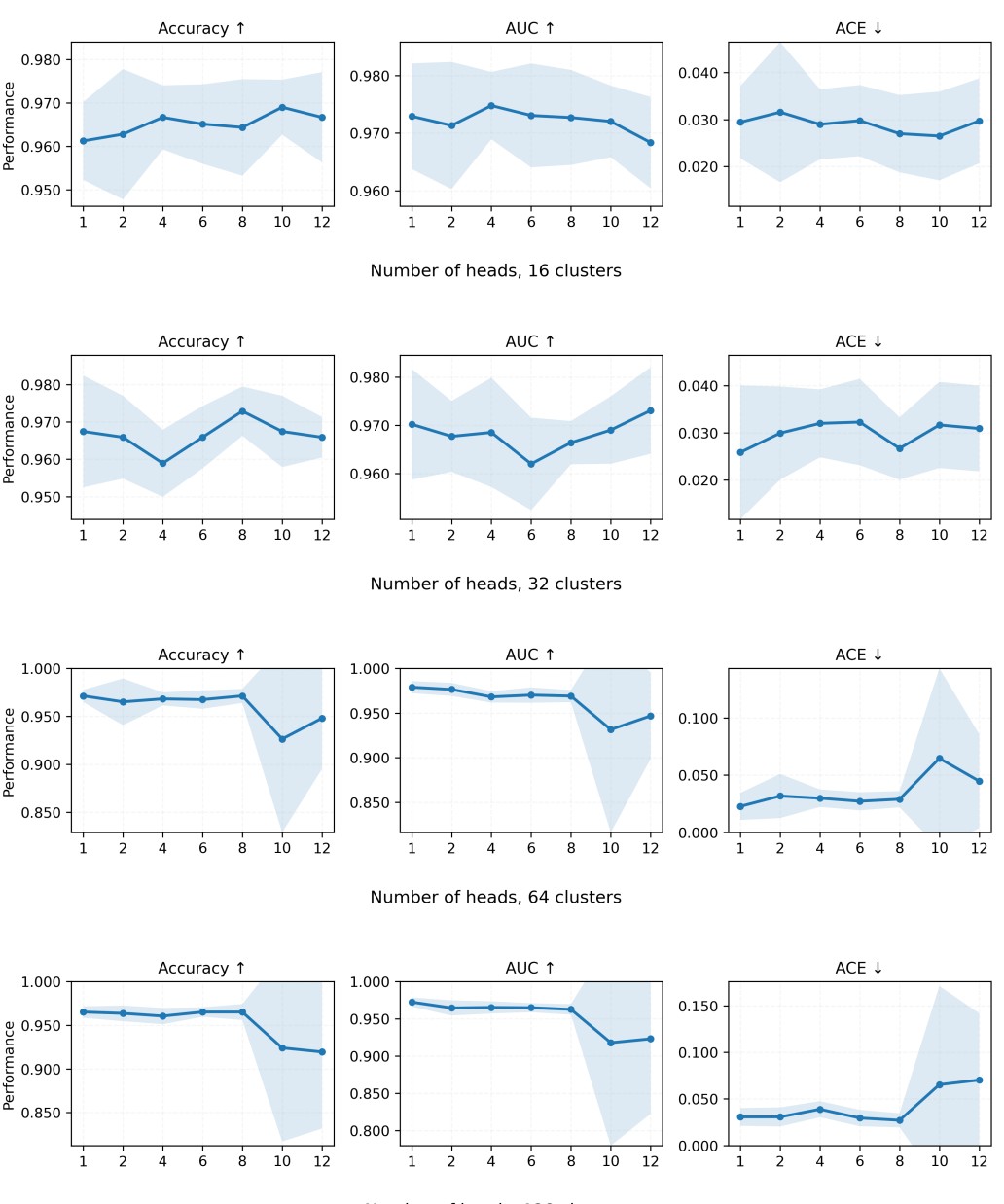

Figure 10: Ablation on the number of attention heads for a single CAPRMIL block on CAMELYON16. For each row, the number of clusters is fixed while the number of heads is varied. From top to bottom: 16, 32, 64, and 128 clusters.

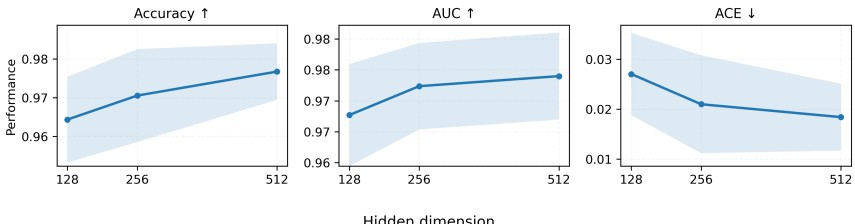

Figure 11: Ablation on the dimensionality of the input projection layer on CAMELYON16. We vary the number of hidden units while keeping the number of clusters (16) and attention heads (8) fixed.

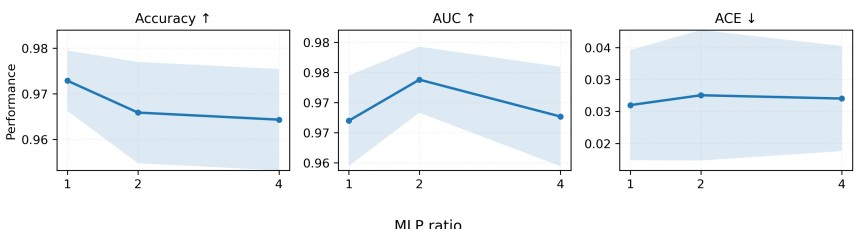

Figure 12: Ablation on the MLP expansion ratio in the CAPRMIL block on CAMELYON16. We vary the expansion factor of the feed-forward network (MLP ratio $\in \{1, 2, 4\}$) while keeping all other components fixed (16 clusters, 8 heads).

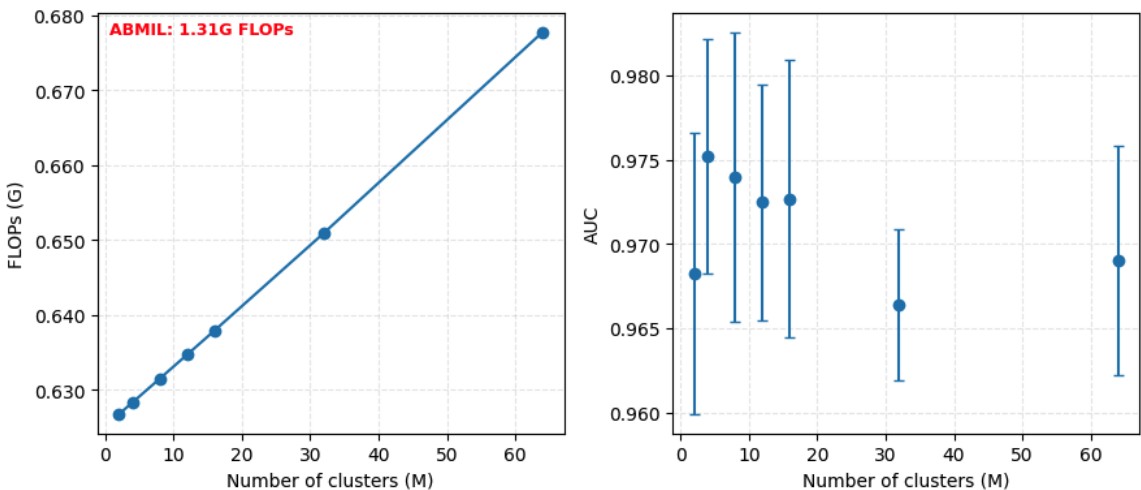

Figure 13: **Effect of the number of clusters $M$ on computational cost and performance. Left:** Inference FLOPs of CAPRMIL as a function of $M$, measured for a bag of 1,000 patch embeddings. The reference FLOPs of ABMIL (1.31G) are shown for comparison. **Right:** CAPRMIL classification performance (mean AUC $\pm$ standard deviation) under 10-fold cross-validation for different values of $M$. FLOPs increase with $M$, while performance saturates beyond small numbers of clusters.

| Init | CAMELYON16 | | TCGA-NSCLC | | PANDA | | BRACS | |
|---|---|---|---|---|---|---|---|---|
| | AUC | ACE | AUC | ACE | $\kappa$ | ACE | AUC | ACE |
| Orthogonal | $0.975_{0.007}$ | $0.028_{0.005}$ | $0.978_{0.016}$ | $0.033_{0.021}$ | $0.953_{0.044}$ | $0.019_{0.023}$ | $0.850_{0.031}$ | $0.189_{0.026}$ |
| Random | $0.964_{0.011}$ | $0.032_{0.012}$ | $0.975_{0.018}$ | $0.029_{0.024}$ | $0.952_{0.046}$ | $0.021_{0.024}$ | $0.851_{0.027}$ | $0.186_{0.020}$ |

Table 6: Ablation of the initialization of $W_{\mathrm{cluster}}$ (slice/cluster projection). Results are reported as mean$_{\mathrm{std}}$ over cross-validation folds for CAMELYON16, TCGA-NSCLC, and PANDA, and over 4 random seeds for BRACS.

| $M$ | PANDA | | BRACS | |
|---|---|---|---|---|
| | $\kappa$ | ACE | AUC | ACE |
| 4 | $0.944_{0.053}$ | $0.021_{0.024}$ | $0.850_{0.031}$ | $0.189_{0.026}$ |
| 6 | $0.945_{0.049}$ | $\mathbf{0.020_{0.022}}$ | $0.852_{0.032}$ | $0.182_{0.019}$ |
| 12 | $\mathbf{0.955_{0.049}}$ | $0.021_{0.026}$ | $0.870_{0.016}$ | $\mathbf{0.153_{0.019}}$ |
| 16 | $0.944_{0.053}$ | $0.021_{0.024}$ | $\mathbf{0.871_{0.024}}$ | $0.183_{0.010}$ |

Table 7: Effect of the number of clusters $M$ on performance and calibration for PANDA and BRACS. Results are reported as mean$_{\mathrm{std}}$ over 5 cross-validation folds for PANDA and 4 seeds for BRACS.

## Appendix C. Pooling operators

Given context-aware patch representations $\mathbf{H}^{(T)} = \{\mathbf{h}_1, \ldots, \mathbf{h}_N\}$ with $\mathbf{h}_n \in \mathbb{R}^D$, we consider the following MIL aggregation operators $\mathcal{A}(\cdot)$:

**Mean pooling.**

$$\mathbf{z}_{\mathrm{mean}} = \frac{1}{N} \sum_{n=1}^{N} \mathbf{h}_n$$

**Max pooling.**

$$\mathbf{z}_{\mathrm{max}} = \max_{n \in \{1, \ldots, N\}} \mathbf{h}_n$$

where the maximum is taken element-wise across instances for each feature.

**Attention pooling.**

$$a_n = \frac{\exp\!\big(\mathbf{w}^\top \tanh(\mathbf{V}\mathbf{h}_n^\top)\big)}{\sum_{m=1}^{N} \exp(\mathbf{w}^\top \tanh(\mathbf{V}\mathbf{h}_m^\top))}, \qquad \mathbf{z}_{\mathrm{attn}} = \sum_{n=1}^{N} a_n \, \mathbf{h}_n$$

**Gated-attention pooling.**

$$a_n = \frac{\exp\!\big(\mathbf{w}^\top \big(\tanh(\mathbf{V}\mathbf{h}_n^\top) \odot \sigma(\mathbf{U}\mathbf{h}_n^\top)\big)\big)}{\sum_{m=1}^{N} \exp(\mathbf{w}^\top \big(\tanh(\mathbf{V}\mathbf{h}_m^\top) \odot \sigma(\mathbf{U}\mathbf{h}_m^\top)\big))}, \qquad \mathbf{z}_{\mathrm{gated}} = \sum_{n=1}^{N} a_n \, \mathbf{h}_n$$

All aggregation operators map $\mathbf{H}^{(T)}$ to a fixed-dimensional slide-level embedding $\mathbf{z} \in \mathbb{R}^D$ and can be used interchangeably without modifying the CAPRMIL blocks.

## Appendix D. Clustering Behavior and Token Specialization

### D.1. Quantitative Analysis of Cluster Utilization

To quantitatively assess cluster utilization and verify that CAPRMIL does not suffer from cluster collapse, we analyze both the uncertainty of patch-to-cluster assignments and the distribution of cluster usage across attention heads. Patch-to-cluster assignment scores are explicitly regularized through three mechanisms: (i) normalization over the $M$ clusters such that $\sum_{m=1}^{M} W_{b,h,n,m} = 1$, where $W_{b,h,n,m}$ denotes the assignment weight of patch $n$ to cluster $m$ for head $h$ in batch $b$; (ii) softmax scaling over the cluster dimension, which sharpens confident assignments; and (iii) a learnable, head-specific temperature parameter that directly controls the entropy of the assignment distribution (Section 3.1.3).

**Assignment entropy.** To quantify whether patches are preferentially routed to specific clusters rather than uniformly distributed, we compute the normalized entropy of the soft cluster assignment vector for each patch and attention head,

$$H(\mathbf{w}_n^{(h)}) = -\frac{1}{\log M} \sum_{m=1}^{M} w_{n,m}^{(h)} \log w_{n,m}^{(h)},$$

where $H = 1$ corresponds to uniform assignment and $H = 0$ to a deterministic assignment. Entropy is computed over 20k randomly sampled patches per dataset (to ensure tractability and reproducibility) and distributions are plotted per head. As shown in Appendix D, Figures 14–15 (violin plots), entropy values remain consistently below 0.5 across heads and datasets, indicating confident and non-uniform patch-to-cluster assignments.

**Cluster occupancy.** Complementarily, we measure per-head cluster occupancy by assigning each patch to its most likely cluster via $\arg\max_m w_{n,m}^{(h)}$. For each slide, we compute the proportion of patches assigned to each cluster per head and then average these proportions across slides. The resulting heatmaps (Appendix D, Figures 14–15) reveal clear head specialization: individual heads predominantly activate distinct subsets of clusters while still utilizing secondary clusters, with this behavior consistent across binary and multiclass datasets.

Together, these entropy and occupancy analyses provide quantitative evidence that CAPRMIL maintains diverse, head-specialized, and non-collapsed cluster assignments, supporting the interpretability and stability of the learned morphology-aware tokens.

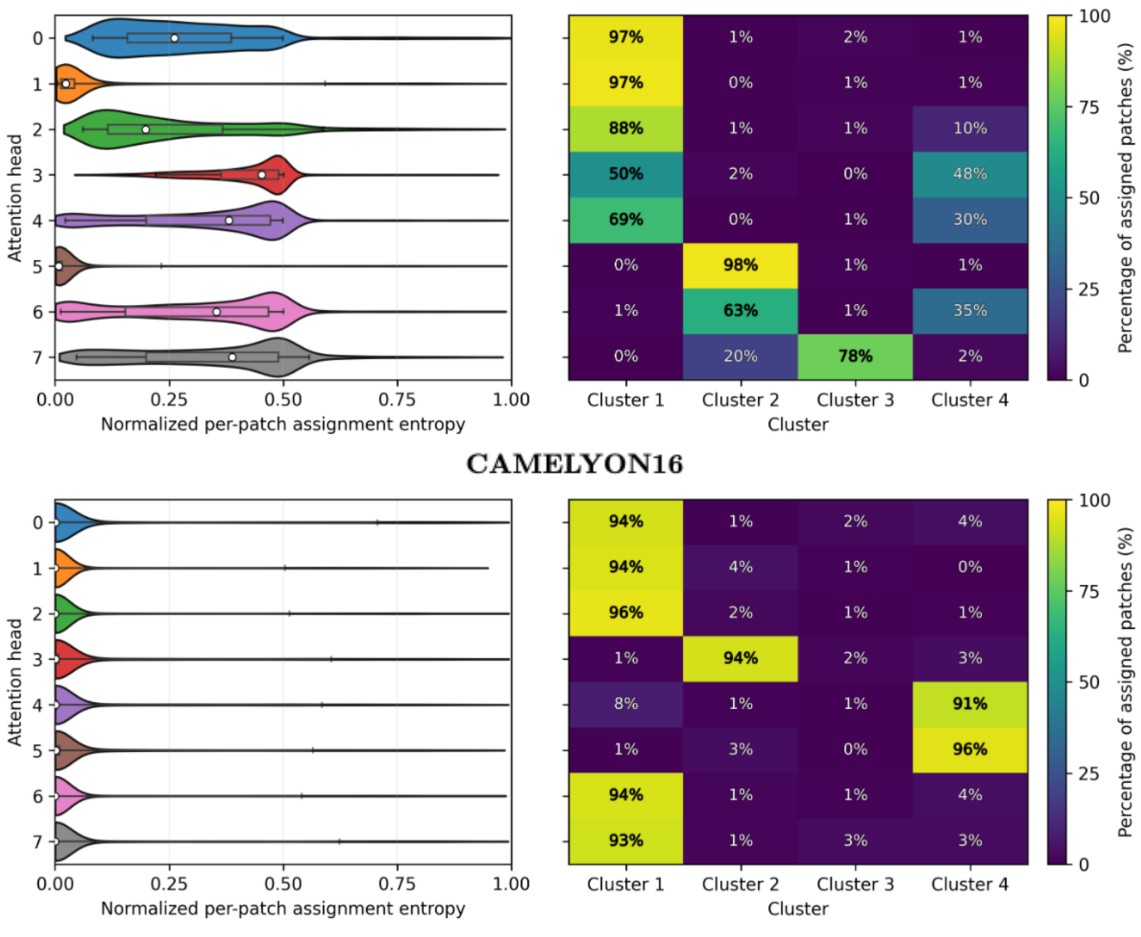

**CAMELYON16**

**TCGA-NSCLC**

Figure 14: **Left:** Distribution of normalized per-patch cluster-assignment entropy for each attention head, computed as $H(\mathbf{w}_n^{(h)}) = -\sum_{m=1}^{M} w_{n,m}^{(h)} \log w_{n,m}^{(h)} / \log M$, where $w_{n,m}^{(h)}$ denotes the soft assignment of patch $n$ to cluster $m$ in head $h$. **Right:** Per-head cluster usage, measured as the percentage of patches assigned to each cluster via hard assignment (argmax over clusters). Across all datasets, entropy distributions remain well below the maximum value, indicating assignment to certain clusters (non-uniform assignment), while cluster usage maps show that heads actively utilize different clusters with distinct specialization patterns rather than collapsing to a single cluster.

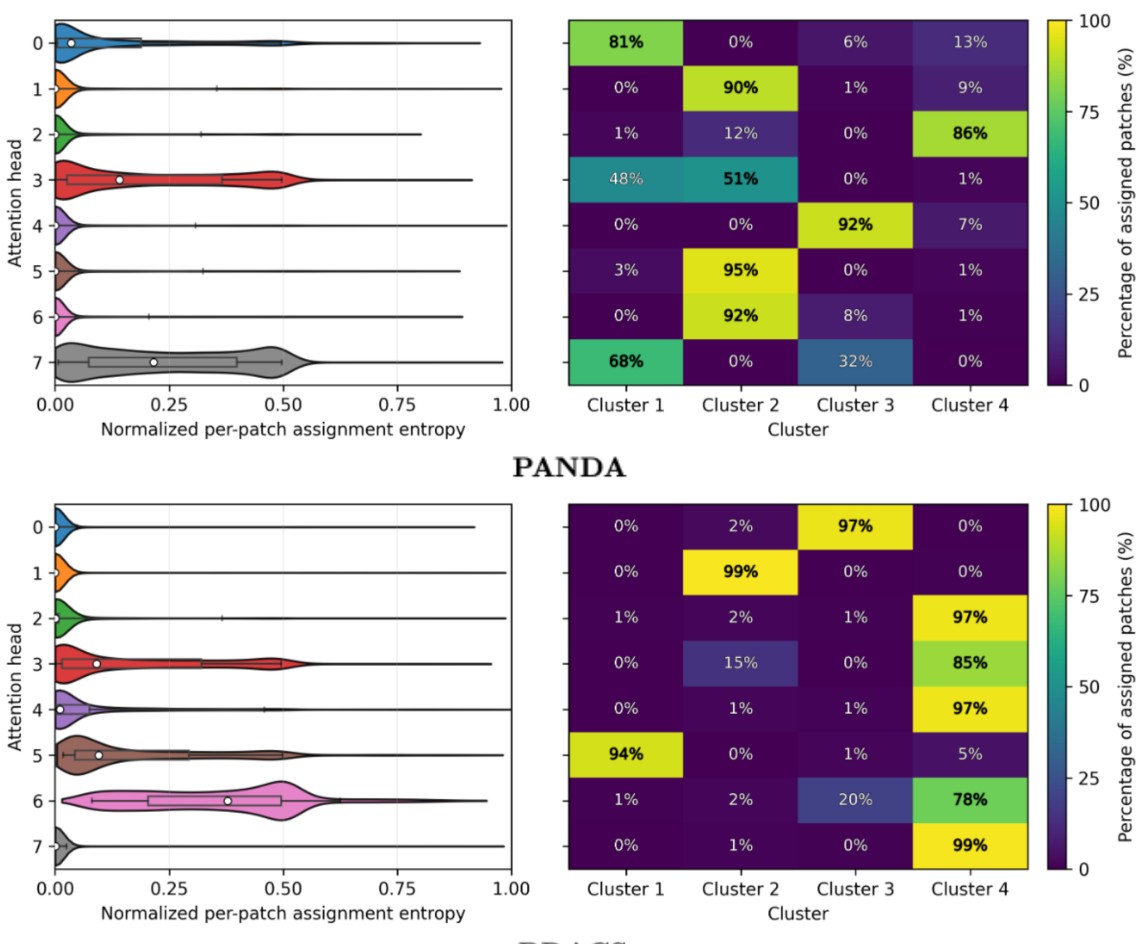

Figure 15: **Left:** Distribution of normalized per-patch cluster-assignment entropy for each attention head, computed as $H(\mathbf{w}_n^{(h)}) = -\sum_{m=1}^{M} w_{n,m}^{(h)} \log w_{n,m}^{(h)} / \log M$, where $w_{n,m}^{(h)}$ denotes the soft assignment of patch $n$ to cluster $m$ in head $h$. **Right:** Per-head cluster usage, measured as the percentage of patches assigned to each cluster via hard assignment (argmax over clusters). Across all datasets, entropy distributions remain well below the maximum value, indicating assignment to certain clusters (non-uniform assignment), while cluster usage maps show that heads actively utilize different clusters with distinct specialization patterns rather than collapsing to a single cluster.

## D.2. Cell-Level Morphological Characterization of Cluster Assignments.

In the absence of direct expert pathologist assessment, we assess the morphological coherence of the learned clusters using two complementary sources of evidence: pixel-level tumor annotations available for CAMELYON16 and cell-level composition analysis using established pretrained models such as HoVerNet. These analyses are reported in **Appendix D, Figures 16–18**.

First, on CAMELYON16, we leverage the available tumor-versus-normal ground-truth masks as a proxy for morphological relevance. By assigning each patch, for one attention head, to the cluster with the highest assignment score (argmax over clusters) and plotting these assignments to slide space, we observe that clusters form spatially coherent regions that align closely with annotated tissue types. As illustrated in **Figure 16** for a representative slide, Cluster 0 predominantly corresponds to normal tissue, Clusters 1 and 3 concentrate on tumor regions, while Cluster 2 is sparsely activated in areas consistent with adipose tissue. This spatial alignment indicates that clusters capture meaningful histological structure rather than arbitrary patch groupings.

Second, to further characterize these regions at a finer scale, we perform a cell-level analysis using HoVerNet on the top 10% highest-scoring patches per cluster. The resulting cell-type distributions and nuclei overlays (**Appendix D, Figures 16–18**) reveal distinct and consistent cellular compositions across clusters: tumor-associated clusters are dominated by neoplastic cells, while others exhibit various proportions of inflammatory, connective, necrotic, or acellular (adipose) tissue. While clusters may partially overlap at tissue boundaries, their cellular composition profiles and visual appearance remain clearly differentiated, indicating that clusters do not collapse onto identical patch sets. Together, these results demonstrate that CAPRMIL clusters correspond to coherent, morphologically meaningful regions and capture biologically interpretable variation in tissue and cellular organization.

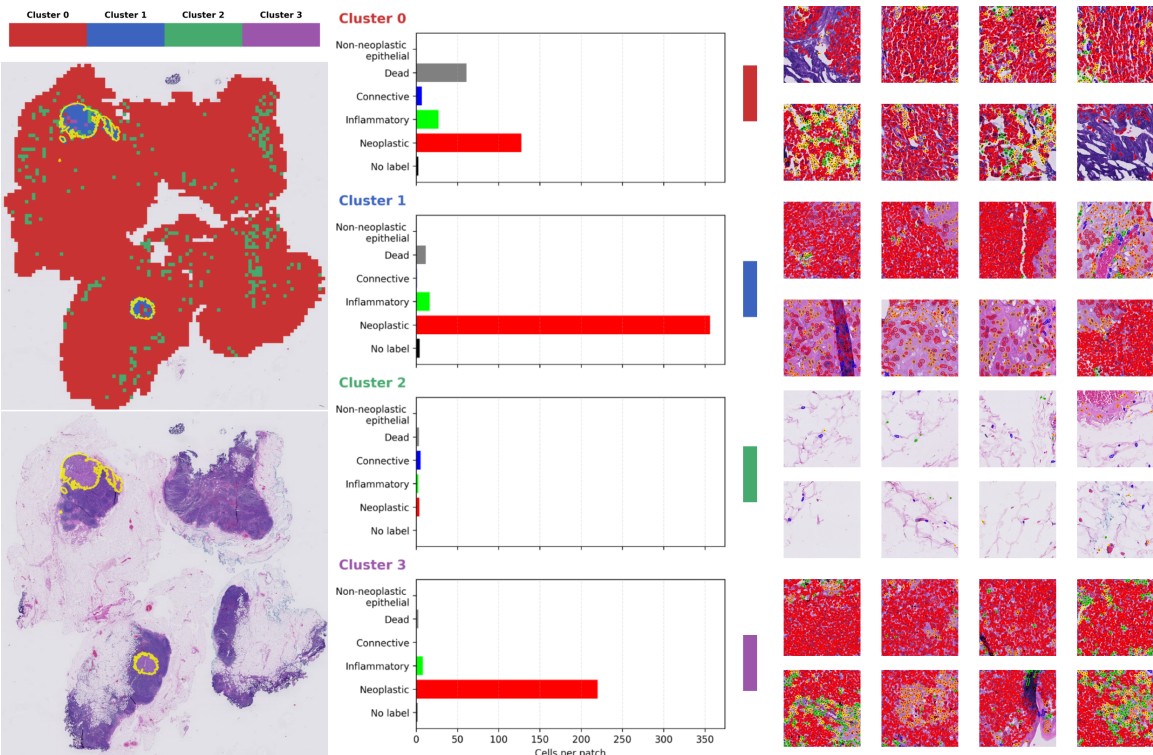

Figure 16: **Cluster-level cellular analysis. Left:** Slide-level cluster maps where each patch is assigned to the cluster with the highest soft-assignment score (argmax over clusters). Yellow contours indicate expert annotations. **Middle:** Cell-type distributions computed by applying HoVer-Net (Graham et al., 2019) to the top 10% highest-confidence patches for each cluster, illustrating cluster-specific cellular composition. **Right:** Representative patches per cluster with HoVer-Net cell segmentation and classification overlays, highlighting characteristic cellular patterns captured by each cluster.

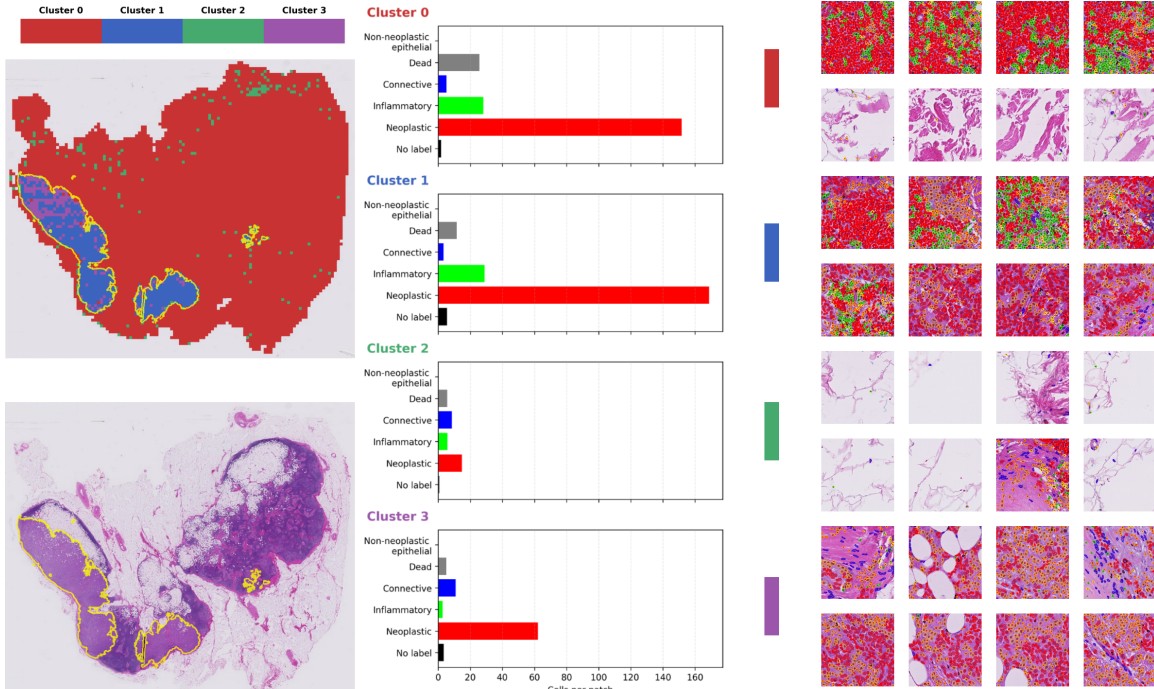

Figure 17: **Cluster-level cellular analysis. Left:** Slide-level cluster maps where each patch is assigned to the cluster with the highest soft-assignment score (argmax over clusters). Yellow contours indicate expert annotations. **Middle:** Cell-type distributions computed by applying HoVer-Net (Graham et al., 2019) to the top 10% highest-confidence patches for each cluster, illustrating cluster-specific cellular composition. **Right:** Representative patches per cluster with HoVer-Net cell segmentation and classification overlays, highlighting characteristic cellular patterns captured by each cluster.

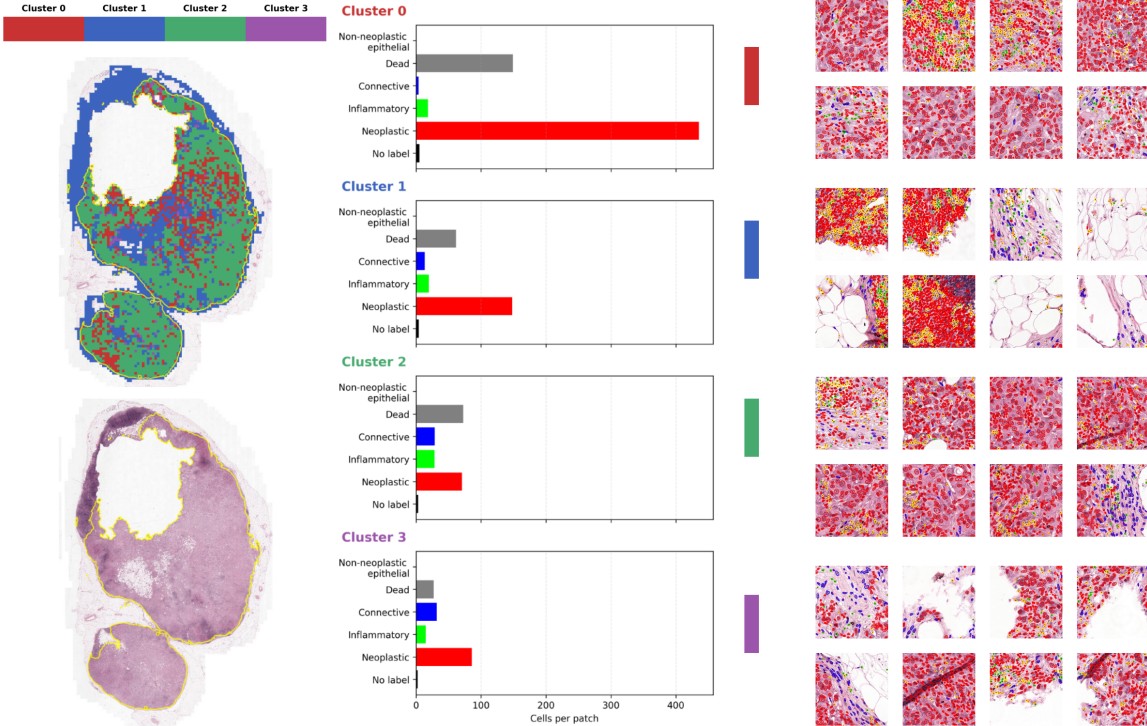

Figure 18: **Cluster-level cellular analysis. Left:** Slide-level cluster maps where each patch is assigned to the cluster with the highest soft-assignment score (argmax over clusters). Yellow contours indicate expert annotations. **Middle:** Cell-type distributions computed by applying HoVer-Net (Graham et al., 2019) to the top 10% highest-confidence patches for each cluster, illustrating cluster-specific cellular composition. **Right:** Representative patches per cluster with HoVer-Net cell segmentation and classification overlays, highlighting characteristic cellular patterns captured by each cluster.

| | Efficiency | | | | | CAMELYON16 | | PANDA | | BRACS | |
|---|---|---|---|---|---|---|---|---|---|---|---|
| Model | Params (k) | FLOPs (G) | Peak GPU (GiB) | Train (s) | Val (s) | AUC | ACE | $\kappa$ | ACE | AUC | ACE |
| Transformer+Mean | 330 | 1.180 | 17.2 | 12.3 | 1.2 | $.977_{.010}$ | $.028_{.012}$ | $.950_{.050}$ | $.022_{.026}$ | − | − |
| CAPRMIL+Mean | 315 | .628 | .263 | 6.2 | 0.8 | $.975_{.006}$ | $.028_{.006}$ | $.944_{.053}$ | $.021_{.024}$ | $.850_{.031}$ | $.189_{.026}$ |

Table 8: Comparison between CAPRMIL+Mean and a parameter-matched (within 5%) full self-attention Transformer+Mean. The full self-attention Transformer could not be evaluated on BRACS due to prohibitive memory requirements caused by large bag sizes (4k–20k).

