# OpenReview forum: "Context-Aware Patch Representations for Multiple Instance Learning"
_MIDL.io/2026/Conference — MIDL 2026 Poster_

### Official Review · Reviewer_RBMQ · 2026-01-09

**Confidence:** 5
**Preliminary Rating:** 3
**Final Rating:** 4

**Summary:**

The paper borrows the “Physics-Attention” pipeline recently proposed for million-node PDE solvers and translates it into a light-weight MIL module for whole-slide images. Instead of attending over N patches, CAPRMIL first soft-clusters patches into M ≪ N morphology tokens, applies self-attention among tokens, and broadcasts the updated tokens back to the patch space. Coupled with a simple mean aggregator, the method matches sota AUC on four public pathology benchmarks while cutting trainable parameters.

**Strengths:**

1. Bring linear-complexity token-attention from neural PDE solvers to gigapixel pathology.
2. Strong empirical efficiency: lower params, FLOPs, memory and training time than TransMIL/DGRMIL with comparable accuracy.
3. Aggregator-agnostic: mean, attention, or gated-attention heads all work, confirming that the context-aware patch features themselves carry most discriminative signal.
4. Interpretability: soft-assignment heat-maps show each token focuses on coherent histological patterns.

**Weaknesses:**

1. The PDE idea is attractive, but there are alread some prototype-based method, which also extract some representative patches to aggregate. It would be better to talk about these methods.
2. Collapse / diversity issue is not analysed—no quantitative measure of cluster usage, entropy, or ablation of orthogonal init.

**Detailed Comments:**

1. Since there are alread some prototype-based method, which also extract some representative patches to aggregate. It would be better to talk about these methods and compare with them.
2. Orthogonal init is mentioned once but never ablated. Provide: (a) train/val cluster-assignment entropy curves; (b) AUC when W_cluster is randomly initialized.
3. Does those selected patches have been checked by physician to make sure they do have morphological pattern?

**Justification Of Final Rating:**

The authors have successfully addressed my concerns. I particularly appreciate the new quantitative analysis on cluster entropy and the ablation study on orthogonal initialization, which confirm the diversity of the learned representations. I am satisfied with the rebuttal and believe this is a solid contribution to the community.

**Justification Of The Preliminary Rating:**

Contribution lies primarily in cross-domain engineering rather than new theory, but the scale of practical gain (≈90 % params/FLOPs reduction) and thorough empirical study make it a useful community tool.

**Questions To Address In The Rebuttal:**

1. Those selected patches, maybe they can be same even for different clusters. Please explain how to prevent this problem.
2. Why is mean pooling + CAPRMIL better than mean pooling + a 2-layer Transformer with the same token bottleneck? (params/FLOPs matched)

---

> ### Author Response · Authors · 2026-01-25
>
> **Effect of Orthogonal vs.\ Random Initialization of $W_{\text{cluster}}$**
>
> We thank the reviewer for highlighting the missing ablation. In CAPRMIL, we follow the design choice of Transolver and initialize the cluster soft assignment matrix $W_{\text{cluster}}$ using an orthogonal initialization, which provides a diverse starting point for the cluster assignment scores. We repeated the experiments using standard random initialization instead. The resulting performance differences are negligible across all evaluated datasets and metrics (**Appendix B, Table 6**). This suggests that CAPRMIL is not strongly dependent on the specific initialization of $W_{\text{cluster}}$, and that orthogonal initialization mainly provides a stable default choice rather than being a critical factor for performance, even on BRACS where we explicitly varied random seeds to assess initialization robustness.
>
> **Comparison to a 2-layer Transformer Baseline.**
>
> CAPRMIL is a one layer Transformer encoder augmented with Physics Attention, applying self-attention over a compact set of $M$ input-conditioned tokens instead of the full set (N), $M \ll N$ (**Section 3.1.3**). To address the reviewer comment,  we compare it to a one layer, parameter-matched (within 5%), Transformer encoder with full self-attention over the $N$ instances and mean pooling. Results in **Appendix D, Table 8** show similar performance and calibration on PANDA, while CAPRMIL is significantly faster and more memory-efficient, with peak GPU utilization reduced by 98% and FLOPs by 55%. Matching FLOPs without altering the task is not possible, since full self-attention scales as $\mathcal{O}(N^2)$, whereas CAPRMIL fundamentally operates at $\mathcal{O}(NM + M^2)$. Subsampling instances to reduce FLOPs would be unfair in MIL, as it may remove diagnostically important regions.

---

> ### Author Response · Authors · 2026-01-25
>
> **Relation to Prototype-Based MIL Methods.**
>
> While our approach is adjacent to prototype-based MIL methods, including SGPMIL, CAPRMIL does not learn a fixed set of input-independent prototype vectors shared across the dataset. Instead, the $M$ bottleneck tokens are input-conditioned latent variables, dynamically computed for each bag via soft clustering and acting as bag-specific summaries. To better position our work, we added a dedicated paragraph in the revised Related Work section (**Prototype-based Multiple Instance Learning**) contrasting CAPRMIL with representative prototype-based MIL methods. Since the reviewer did not point to any particular citation, we hope the discussed methods address the question; however, we will be happy to include additional methods during the discussion period if any were missed. Moreover, we added in the main paper additional comparisons with the prototype-based PAMIL method (**Table 1, Figure 4, and Appendix B, Table 5**). Our method is consistently better while reducing parameters by $60$%, FLOPs by $52$%, inference time by 0.1s, and improving AUC by 2% in BRACS.
>
> **Quantitative Analysis of Cluster Utilization.**
>
> Patch-to-cluster assignments are explicitly regularized to avoid collapse via assignment score normalization over clusters, softmax scaling, and a learnable head-specific temperature that controls assignment entropy (**Section 3.1.3, Soft Clustering**). For each attention head, patches are softly assigned to clusters via normalized assignment scores, allowing each patch to contribute to multiple clusters with varying confidence. To further evaluate this, we quantify cluster usage using two complementary analyses (**Appendix D.1, Figures 14–15**). First, the normalized entropy of the soft cluster assignments remains consistently below 0.5 across datasets and heads, indicating preferential rather than uniform assignment. Second, per-head cluster occupancy heatmaps show clear head specialization, with different heads focusing on distinct subsets of clusters and feature subspaces, while some exhibit more distributed behavior depending on the input, consistently across datasets. A detailed description of how assignment entropy and cluster occupancy are computed is provided in **Appendix D.1: Quantitative Analysis of Cluster Utilization**. Together, these analyses provide quantitative evidence that CAPRMIL avoids cluster collapse and maintains meaningful, diverse, and head-specialized patch-to-cluster assignments.
>
> **Cell-Level Morphological Characterization of Cluster Assignments.**
>
> To assess the morphological coherence of learned clusters, we provide two complementary forms of evidence (**Appendix D, Figures 16–18**). On CAMELYON16, cluster assignments form spatially coherent regions that align with ground-truth pixel-level tumor annotations. Cluster maps are obtained by assigning each patch to its highest-scoring cluster for one attention head. We further perform a cell-level analysis using HoVerNet on high-confidence patches per cluster, revealing distinct and consistent cellular compositions across clusters: clusters associated with tumor regions are dominated by neoplastic cells, while others exhibit different proportions of inflammatory, connective, necrotic, or acellular (adipose) tissue. While overlap may occur at tissue boundaries, their spatial organization and cellular profiles remain clearly differentiated, indicating that clusters do not collapse onto identical patch sets. A detailed description of the experimental setup and analysis protocol is provided in the revised manuscript (**Appendix D.2: Cell-Level Morphological Characterization of Cluster Assignments**).

---

### Official Review · Reviewer_oS9a · 2026-01-10

**Confidence:** 4
**Preliminary Rating:** 3
**Final Rating:** 4

**Summary:**

The authors propose a new MIL method for aggregating patch-level feature vectors into a slide-level prediction in computational pathology.

The proposed CAPRMIL refines the original patch-level features into context-aware patch-level feature vectors, which then can be combined with a simple mean pooling aggregator (CAPRMIL can also be combined with ABMIL or other trainable aggregators).

CAPRMIL achieves competitive performance on four classification datasets compared to ABMIL and other MIL methods, while reducing the number of trainable parameters and FLOPs. However, CAPRMIL is slightly slower and has a slightly higher peak GPU memory usage than ABMIL in practice.

**Strengths:**

- The proposed CAPRMIL approach conceptually makes sense overall.

- CAPRMIL with mean pooling aggregation reduces the number of parameters and FLOPs compared to ABMIL and other MIL methods, while achieving competitive performance on four classification datasets.

**Weaknesses:**

- The paper contains quite a few typos and similar small issues, and could definitely be a bit more well written overall. In particular, Section 3.1.3 is a bit convoluted and difficult to follow, especially the "Soft Clustering" paragraph.

- Although CAPRMIL with mean pooling aggregation reduces the number of parameters and FLOPs compared to ABMIL, it is slightly slower and has higher GPU memory consumption in practice (according to Figure 4). Given that CAPRMIL and ABMIL also have similar classification performance, the practical benefit of CAPRMIL is somewhat unclear.

**Detailed Comments:**

Minor things:
- In Figure 4, "TSMIL" --> "CAPRMIL"?
- This is a very minor thing, but why the upper-case S in "Solvers" in the abstract and Section 1?
- Section 2, "probabilistic-based MIL such as the one introduced at Cui et al. (Cui et al., 2022) argued that standard attention scores are unreliable proxies for interpretability" --> "probabilistic-based MIL methods, such as the one introduced by Cui et al. (2022), argue that standard attention scores are unreliable proxies for interpretability" or similar?
- Section 2, "Similarly, Lolos et al. targeted the lack of uncertainty estimation in deterministic models and introduced SGPMIL (Lolos et al., 2025)" --> "Similarly, Lolos et al. (2025) targeted the lack of uncertainty estimation in deterministic models and introduced SGPMIL", perhaps?
- Similar citation formatting tweaks could also be made in the "The Transolver Architecture" paragraph of Section 2.
- Section 3, "we propose CAPRMIL , a novel" --> "we propose CAPRMIL, a novel".
- Start of 3.1.1, "A WSI is represented" --> "A batch of WSIs is represented"?
- End of 3.1.2, "CAPRMIL attention returns the concatenated output of all heads": don't quite understand what you mean here.
- Figure 2 caption, "Each CAPRMIL Block contains H CAPRMIL Attention heads and their output is concatenated" --> "Each CAPRMIL Block contains H CAPRMIL Attention heads and their outputs are concatenated"?
- 3.1.3, "broadcasting the transited tokens back to the input space": perhaps use updated/refined/transformed instead of "transited"?
- Section 3.1.4: Might want to describe the general case here instead, to follow Figure 1 (you use other aggregation methods than mean pooling as well)?
- 4.1, "As seen in Figures 3 and A 5 - 7": typo?
- 4.2, "leading competitors , while operating" --> "leading competitors, while operating".
- 4.3, "Table A 3" --> "Table A3". You should probably also change "Table 3" to "Table A3" in the appendix.

**Justification Of Final Rating:**

The other reviewers are positive overall, and the authors provided a solid rebuttal that addressed some of my concerns. While the practical benefit compared to ABMIL is still a bit unclear, the proposed CAPRMIL does make sense overall and can probably be a useful alternative method.

**Justification Of The Preliminary Rating:**

The proposed CAPRMIL approach makes sense overall and achieves competitive performance compared to ABMIL, but it is not entirely clear to me what the practical benefits actually are (why should a practitioner choose to use CAPRMIL instead of ABMIL?). Also, the paper could definitely be a bit more well written overall. Therefore, I am currently borderline on this paper, leaning slightly towards reject.

**Questions To Address In The Rebuttal:**

- Why should a practitioner choose CAPRMIL with mean pooling over ABMIL? What are the main practical benefits?

- Could you update Section 3.1.3 and address various minor issues to improve the presentation?

---

> ### Author Response · Authors · 2026-01-25
>
> **Clarification of Section~3.1.3 and Presentation Revisions.**
>
> We thank the reviewer for the detailed feedback on the presentation and apologize for the lack of clarity of the original manuscript. In response, we have revised **Section 3.1.3** to improve readability and better explain the soft clustering mechanism. The section has been streamlined, terminology has been clarified, and ambiguous phrasing (e.g., ''transited tokens'') has been replaced with clearer descriptions (e.g., ''refined'' or ''updated'' tokens). We also expanded the explanation to more clearly convey how token refinement and broadcasting operate within the CAPRMIL block, and how this design relates to **Figure 1**.
>
> In addition, we carefully addressed the reviewer’s minor comments and typographical issues throughout the manuscript. This includes correcting naming inconsistencies (e.g., ''TSMIL'' to ''CAPRMIL'' in **Figure 4**), standardizing citation formatting, fixing grammatical and punctuation errors, and improving figure captions and section phrasing for clarity. We also updated **Section 3.1.4** to describe the general aggregation setting, consistent with the architecture shown in **Figure 1**, and added pooling operator formulas in the appendix (**Appendix C**).
>
> We believe these revisions substantially improve the clarity, correctness, and overall readability of the manuscript. All these changes are highlighted in red in the main manuscript.
>
>
> **Practical Benefits of CAPRMIL vs. ABMIL.**
>
> We thank the reviewer for this important question. While CAPRMIL and ABMIL achieve comparable performance when trained on the full dataset, CAPRMIL is designed with a different practical objective: to improve representation learning under a multi-instance learning setting while keeping aggregation lightweight and modular. Although CAPRMIL exhibits a slightly higher peak GPU memory usage and marginally slower training time than ABMIL (**Figure 4**), this overhead is modest and confined to the construction of context-aware patch representations, achieved via soft clustering of patch embeddings into a small set of morphology-aware tokens, followed by token-level self-attention and context broadcasting back to the patch space. Crucially, CAPRMIL remains substantially more efficient overall, reducing inference FLOPs by approximately $52$% and parameters by $48$% compared to ABMIL (**Table 1**). This advantage is further reflected in generalization performance in low-data regimes added for the revision: as shown in main manuscript **Table 2**, when training on only $10$%, $25$%, and $50$% of PANDA, CAPRMIL consistently outperforms ABMIL with absolute gains of up to $3.5$%, $4.7$%, and $4.3$% in accuracy, and $1.9$%, $2.0$%, and $2.7$% in $\kappa$, exhibiting better ACE across all regimes.
>
> From a practical standpoint, these results indicate that CAPRMIL provides a more efficient alternative to ABMIL, achieving comparable or improved performance while substantially reducing parameter count and computational cost. This efficiency translates into more stable training and improved generalization in low-data settings, as observed in PANDA, clarifying the practical motivation for choosing CAPRMIL over ABMIL.

---

> > ### Comment · Reviewer_oS9a · 2026-01-31
> >
> > Thank you for the response.
> >
> > I have read the other reviews and all author responses.
> >
> > The other reviewers are positive overall, and the authors provided a solid rebuttal that addressed some of my concerns. While the practical benefit compared to ABMIL is still a bit unclear, the proposed CAPRMIL does make sense overall and can probably be a useful alternative method.
> >
> > Therefore, I have raised my score to "4: Weak accept".

---

> > > ### Author Response · Authors · 2026-01-31
> > >
> > > Thank you for your thorough review, positive assessment and recognition of CAPRMIL as a useful alternative approach. We appreciate the thoughtful consideration and are pleased that the rebuttal and revisions helped clarify the method and address several of the concerns raised.

---

### Official Review · Reviewer_6SUd · 2026-01-10

**Confidence:** 5
**Preliminary Rating:** 5
**Final Rating:** 5

**Summary:**

The paper proposes a method to address the computational complexity of using self-attention as part of a multiple-instance learning (MIL) frameworks in computational pathology. The methodology is inspired from Transolver, which is a recently proposed surrogate PDE solver. The authors made an observation that the problems of exploding quadratic complexity are present in both PDE solvers that operate on a large number of mesh points and in MIL methods in computational pathology that operate on a large number of patches extracted from whole slide images. In this work they adapt the Transolver method, which clusters individual instances, applies self-attention to the cluster representations and then back projects, to MIL for computational pathology.

**Strengths:**

- The paper is well-written with good introduction of related work and motivation for the current work and problem that it addresses.
- Generally, I really appreciate approaches that bring ideas and methodology from one application domain to another. While this might also be seen as a weakness in the sense of reduced novelty, I do not consider it to be.
- The work is well executed with large and robust set of experiments, including and ablation study.
- The proposed methodology is interesting and relevant and can have impact on the computational pathology field

**Weaknesses:**

- The number of clusters used is only 4 in all experiments. This was set based on the ablation study that showed stable performance. From what I can tell, the ablation study was done on the Camelyon dataset which is a visual "simple" tasks where I can imagine 4 clusters might be sufficient. I wonder if the authors left some performance on the table by using such small M for all tasks.
- Since the computational complexity depends on the number of clusters, it is unclear how much increasing M will contribute to increased computational complexity.

**Detailed Comments:**

See my comments above.

**Justification Of Final Rating:**

The authors have addressed my concerns, which were minor to begin with. Concerns raised by other reviewers were also addressed. The new experiments further strengthened the manuscript. Because of this I keep my original recommendation.

**Justification Of The Preliminary Rating:**

It is a well-written and executed paper that while only "transplants" an idea from a different application domain, I still find to be quite relevant and interesting for the computational pathology community.

**Questions To Address In The Rebuttal:**

I would like to see experiments with larger value for M for the other datasets. Also, it would be good to include max in addition to mean pooling to the baseline as this can have very good performance for some of the tasks.

---

> ### Author Response · Authors · 2026-01-25
>
> **Effect of the Number of Clusters ($M$).**
>
> We thank the reviewer for the thoughtful comment regarding the choice of the number of clusters $M$. Although the ablation study that is presented at **Appendix A, Table 3 and Appendix B, Figure 8** is focused on CAMELYON16, we additionally evaluated the effect of increasing $M$ on more visually and semantically complex multiclass datasets, specifically PANDA (6 classes) and BRACS (3 classes). These extra experiments have been added to **Appendix B, Table 7** in the updated version of our paper.
> As shown in **Appendix B, Table 7**, increasing the number of clusters from $M=4$ to $M=6,12$ and $16$ results in only small changes in performance, with no consistent or statistically significant improvements across AUC/Kappa, and calibration metrics. In practice, performance tends to saturate at relatively low values of $M$. For example, we observe that two clusters are sufficient for binary tasks such as CAMELYON16, while approximately four clusters are adequate for multiclass settings, beyond which additional clusters provide limited benefit. Since the clusters are trained together with the final task, a small number of classes could be addressed with a small number of clusters. Particularly, in PANDA, a task with 6 classes, performance gains by increasing the number of clusters are within standard deviation to the baseline (4 clusters).
> Overall, these findings suggest that while increasing $M$ can lead to slight performance variations, larger numbers of clusters do not necessarily yield increased discriminative power. A small set of morphology-aware tokens already captures the dominant histological patterns relevant for slide-level prediction (**Appendix D.2, Figures 16-18**), and this behavior is consistent across datasets with different numbers of classes and varying visual complexity. From a practical perspective, this saturation effect highlights an important trade-off in CAPRMIL. Using a small $M$ maintains linear computational complexity and efficiency while achieving competitive performance. Increasing $M$ would increase computational cost without clear gains, even in multiclass cases, which motivated our use of $M=4$ as a stable and effective default throughout the experiments, for problems with up to 6 classes.
>
> Finally, to address potential concerns about under-utilization or cluster collapse when using small $M$ (as reviewer [RBMQ] asked), we include additional analyses in the appendix (**Appendix D.1, Figures 14-15**),  that report cluster usage statistics and per-patch assignment entropy across heads and datasets. These results indicate that clusters are actively and diversely used, and that patch-to-cluster assignments retain meaningful uncertainty rather than collapsing to a degenerate solution. Detailed computation information for these figures is included in their respective captions.
>
> **Computational Impact of Increasing $M$.**
>
> We evaluated how increasing $M$ affects both model size and computational cost (**Appendix B, Figure 13**). In practice, the impact is modest. Increasing the number of clusters from $M=4$ to $M=64$ results in a negligible increase in the number of trainable parameters, from approximately $314$K to $315$K. Similarly, the increase in FLOPs remains limited, from $626$M to $677$M. Since CAPRMIL applies self-attention in a reduced token space rather than directly over all patch embeddings, the overall computational cost grows with $M^2$. However, $M$ is small in practice even for a large number of classes. Even at $M=64$, the FLOPs increase by only about $7$% compared to $M=2$, and remain substantially lower than those of the closest competitor, ABMIL, which operates attention directly in the patch space.
> These results indicate that while increasing $M$ does introduce additional computation, the overhead is well controlled and does not negate the efficiency benefits of CAPRMIL. This further supports our design choice of using a small number of clusters to balance representational capacity and computational efficiency. At the same time, our approach remains modular and can be used with multiple different pooling operators.
>
> **Max Pooling Baseline.**
>
> We thank the reviewer for this suggestion. We have added a max-pooling baseline to Table 3 in the main paper and evaluated it across all datasets. As shown in the updated results, CAPRMIL combined with max pooling yields a small but consistent performance improvement on CAMELYON16, PANDA, and BRACS compared to mean pooling, while maintaining the same parameter count.
> These results confirm that max pooling can be a competitive aggregation strategy for certain tasks. The overall performance trends remain consistent across aggregation methods, supporting our claim that the primary benefit of CAPRMIL lies in learning improved context-aware patch representations, rather than relying on a specific pooling operator.

---

### Author Rebuttal · Authors · 2026-01-25

**Rebuttal:**

We thank the reviewers for their thoughtful, thorough, and constructive feedback, and for the overall positive assessment of our work including (i) the positive  evaluation of our experimental study and ablations [6SUd,oS9a, RBMQ]. (ii) the interest and relevance of our work [6SUd], and (iii) the positive assessment for the aggregator-agnostic, and interpretable design of our method [RBMQ]. Below, we provide point-by-point responses to each reviewer’s comments and indicate the corresponding modifications made to the manuscript. Our revision includes: (i) extra ablations requested by the reviewers, (ii) extra comparisons with baselines and competitive methods, (iii) extensive revisions on the main paper including extra discussion and presentation of the contributions of our method.  All changes in the revised PDF are highlighted in red. We believe that these revisions have strengthened the paper’s contributions, improved clarity and reproducibility.

**Supporting Material:**

/attachment/7489992de855af8d3c6873de69fa27b8196ae3f4.pdf

---

### Meta-Review · Area_Chair_ryDj · 2026-02-08

**Recommendation:** Accept (Poster)
**Confidence:** 4

**Metareview:**

All reviewers agree that the proposed method is novel and effective, with solid experiments, demonstrating significant practical benefits in computational pathology.

---

### Decision · Program_Chairs · 2026-02-13

Accept (Poster)